# PYRAMIDAL PATCHIFICATION FLOW FOR VISUAL GENERATION

**Hui Li** [1]**, Baoyou Chen** [3]**, Liwei Zhang** [3]**, Jiaye Li** [1]**,**
**Jingdong Wang**[2]**, Siyu Zhu** [1,3,4]
[1]Fudan University, [2]Baidu Inc.
[3]Shanghai Innovation Institute
[4]Shanghai Academy of AI for Science

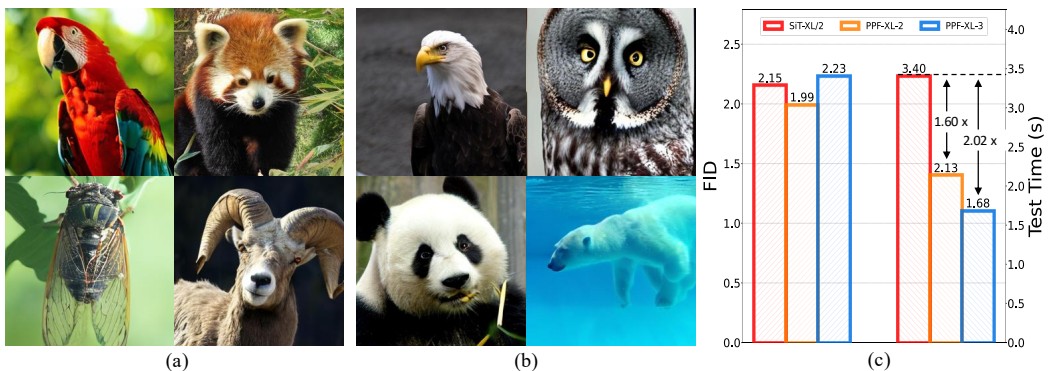

Figure 1: Pyramidal Patchification Flow (PPFlow) achieves state-of-the-art image generation quality with accelerated denoising processes. (a) and (b) show visual samples from two of our class-conditional PPF-XL-2 and PPF-XL-3 trained on ImageNet. (c) indicates that PPF-XL-2 and PPF-XL-3 obtains $1.6 \times$ and $2.0 \times$ inference acceleration with comparable FID scores.

## ABSTRACT

Diffusion Transformers (DiTs) typically use the same patch size for $\mathrm{Patchify}$ across timesteps, enforcing a constant token budget across timesteps. In this paper, we introduce Pyramidal Patchification Flow (PPFlow), which reduces the number of tokens for high-noise timesteps to improve the sampling efficiency. The idea is simple: use larger patches at higher-noise timesteps and smaller patches at lower-noise timesteps. The implementation is easy: share the DiT's transformer blocks across timesteps, and learn separate linear projections for different patch sizes in $\mathrm{Patchify}$ and $\mathrm{Unpatchify}$. Unlike Pyramidal Flow that operates on pyramid representations, our approach operates over full latent representations, eliminating trajectory jump points, and thus avoiding re-noising tricks for sampling. Training from pretrained SiT-XL/2 requires only $+8.9\%$ additional training FLOPs and delivers $2.02\times$ denoising speedups with image generation quality kept; training from scratch achieves comparable sampling speedup, e.g., $2.04\times$ speedup in SiT-B. Training from text-to-image model FLUX.1, PPFlow can achieve $1.61 - 1.86\times$ speedup from 512 to 2048 resolution with comparable quality. The code and checkpoint are at https://github.com/fudan-generative-vision/PPFlow.

## 1 INTRODUCTION

Diffusion (Ho et al., 2020; Song & Ermon, 2019; Song et al., 2021b; Rombach et al., 2022; Saharia et al., 2022b) and flow-based models (Papamakarios et al., 2021; Xu et al., 2022; Liu et al., 2023b; Lipman et al., 2022; Esser et al., 2024) set the state of the art in visual generation. They comprise a noising process that maps data to noise and a denoising process that iteratively evaluates a learned

network to transport a sample from Gaussian noise to the data distribution. While highly effective, the denoising trajectory typically requires many expensive network evaluations.

A large body of work reduces denoising cost mainly along three lines: (i) reducing the number of function evaluations (e.g., DDIM (Song et al., 2021a), distillation (Salimans & Ho, 2022; Meng et al., 2023b; Yin et al., 2024a), consistency models (Song et al., 2023; Luo et al., 2023), and one-step diffusion (Yin et al., 2024b; Liu et al., 2023c; Frans et al., 2024)); (ii) lowering the cost of each function evaluation via model compression and architectural choices, including quantization (He et al., 2023; Fang et al., 2023; Zhao et al., 2024; Li et al., 2023; Huang et al., 2024), pruning (Xi et al., 2025; Xia et al., 2025; Zhang et al., 2025a;b; Xie et al., 2024), and reducing token counts in denoising process using coarse representations or cascaded designs (Ho et al., 2022b; Saharia et al., 2022a; Ho et al., 2022a; Saharia et al., 2022c; Pernias et al., 2023; Gu et al., 2023; Atzmon et al., 2024; Jin et al., 2024; Chen et al., 2025c); and (iii) other ways such as removing or amortizing classifier-free guidance (Ho & Salimans, 2021; Fan et al., 2025; Chen et al., 2025a; Meng et al., 2023a). Among these, approaches that vary spatial resolution over time (e.g., pyramidal or cascaded generation) reduce tokens at early, noisier timesteps but can introduce resolution "jumps", which complicate training and inference and break trajectory continuity.

The interest of this paper lies in improving the denoising network evaluation efficiency with a focus on reducing the number of tokens input to DiT blocks. We present a simple and easily-implemented approach, Pyramidal Patchification Flow (PPFlow). Diffusion Transformer (DiT) Peebles & Xie (2023); Ma et al. (2024); Esser et al. (2024) exploits a $\mathtt{patchify}$ operation to control the number of tokens and accordingly the computation complexity. It applies the same patch size, typically, $2 \times 2$ for all the time steps. Our approach introduces simple modifications. It adopts a pyramid patchification scheme. The patch size is larger for timesteps with higher noise. The patch sizes of a two-pyramid-level example are: $4 \times 4$ and $2 \times 2$. Each level has its own parameters for linear projections mapping patch representations to token representations in $\mathrm{Patchify}$. Similarly, each level has its own linear projection parameters in $\mathrm{Unpatchify}$. All the levels adopt the same parameters in DiT blocks.

Our approach is related to and clearly different from the recently-developed Pyramidal Flow Jin et al. (2024) and PixelFlow Chen et al. (2025c). The similarity lies in that the number of tokens at the timesteps with higher noise is smaller. The differences include: (i) Our approach operates over full-resolution latent representations; Pyramidal Flow operates over pyramid representations. This is illustrated in Figure 2. (ii) Our approach still satisfies the Continuity Equation. Pyramidal Flow Jin et al. (2024) indicates that it does not satisfy the equation as the representation resolution varies along the trajectory. (iii) Our approach has no issue of "jump points" Campbell et al. (2023b), and the inference is the same as normal DiT. Pyramidal Flow adopts a carefully-designed renoising trick.

We evaluate PPFlow in two training regimes: training from scratch and adaptation from pretrained DiTs. From scratch, two- and three-stage PPFlow achieve comparable or better quality than SiT-B/2, with FID 3.83 and 4.43 versus 4.46, while yielding $1.61\times$ and $2.04\times$ denoising speedups. When initialized from pretrained DiTs (e.g., SiT-XL/2), PPFlow requires only $8.9\%$ and $7.1\%$ additional training FLOPs for two- and three-stage variants, maintains similar generation quality, and delivers $1.60\times$ and $2.02\times$ inference speedups, respectively. Qualitative examples are shown in Figure 1. Training from text-to-image model FLUX.1, PPFlow can achieve $1.61 - 1.86\times$ speedup from 512 to 2048 resolution with comparable quality.

## 2 RELATED WORK

**Reducing the number of function evaluations.** The early algorithm, e.g., DDIM (Song et al., 2021a), greatly reduces the number of function evaluation. Distillation techniques are also widely-studied (Salimans & Ho, 2022; Meng et al., 2023b; Yin et al., 2024a). Consistency models (Song et al., 2023; Luo et al., 2023) distill pre-trained diffusion models to models with a small number of sampling steps, including multi-step and single-step sampling. Recently, various one-step diffusion frameworks (Yin et al., 2024b; Liu et al., 2023c; Frans et al., 2024) have been developed.

**Reducing the cost of the function evaluation.** Model quantization is applied to diffusion/flow-based models for faster inference, including post-training or quantization-aware training (He et al.,

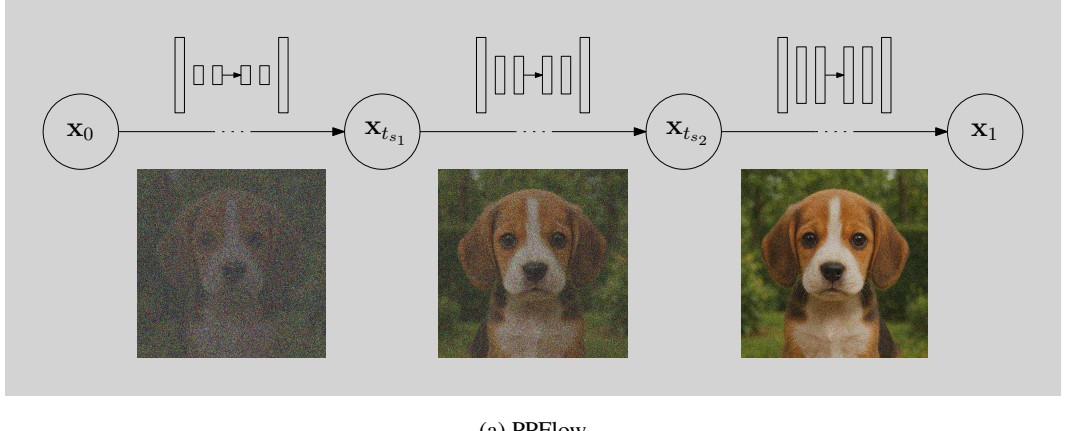

(a) PPFlow

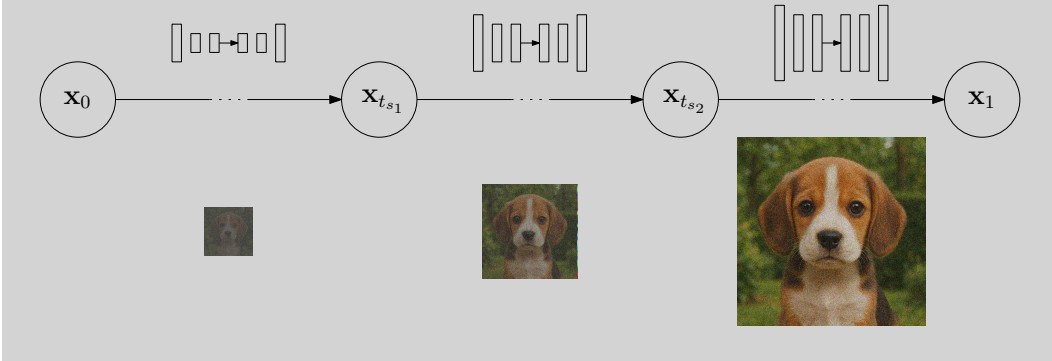

(b) Pyramidal Flow (Jin et al., 2024)

Figure 2: Conceptual comparison. (a) A three-level PPFlow example. The patch sizes in $\mathrm{Patchify}$ are larger for higher-noise timesteps and smaller for lower-noise timesteps. The representation resolutions for all the three levels are the same and full. (b) Pyramidal Flow Jin et al. (2024). We illustrate it for image generation. It operates over pyramid representations: smaller representation resolution for higher noise and larger representation resolution for lower noise.

2023; Fang et al., 2023; Li et al., 2023; Huang et al., 2024). Structural and video-specific quantization methods are studied in (Zhao et al., 2024).

**Multi-scale and cascaded generation.** Multi-scale generation progressively generate images or latent representations from low resolution to high resolution (Ho et al., 2022b; Saharia et al., 2022a; Ho et al., 2022a; Saharia et al., 2022c; Pernias et al., 2023; Gu et al., 2023; Atzmon et al., 2024; Jin et al., 2024; Chen et al., 2025c). Pyramidal flow (Jin et al., 2024), closely related to our approach, reinterprets the denoising trajectory as a series of pyramid stages, where only the final stage operates at the full resolution. The inference introduces a renoising trick to carefully handle jump points (Campbell et al., 2023a), i.e., latent representation resolution change, across stages, ensuring continuity of the probability path. Our approach operates over full-resolution representations, and does not require such a renoising trick.

**Patchification with varying patch sizes.** Training one ViT model with varying patch sizes is studied in FlexiViT (Beyer et al., 2023). It is extended to train one diffusion/flow matching model with varying patch sizes (in implementation, a few parameters in the model are dedicated to each patch size) in the concurrent works, FlexiDiT (Anagnostidis et al., 2025) and Lumina-Video (Liu et al., 2025). The denoising process in FlexiDiT and Lumina-Video, is similar to ours: larger patch sizes for higher-noise time steps, and smaller patch sizes for lower-noise time steps. The training process is different from our approach: They train the model of each patch size for all the timesteps of all the noise degrees, optionally with more lower-noise timesteps for training the model with a

smaller patch size and more higher-noise timesteps for training the model with a larger patch size in Lumina-Video. This difference leads to the inconsistency between the training and testing processes for FlexiDiT and Lumina-Video. One benefit from our approach training the model of each patch size for the range of the corresponding timesteps is that the model of a patch size can better handle the corresponding specific noise degrees. This benefit is verified by our empirical results.

# 3 METHOD

## 3.1 PRELIMINARIES: FLOW MATCHING AND DiT PATCHIFICATION

**Flow matching.** Flow-based generative models (Papamakarios et al., 2021; Xu et al., 2022; Liu et al., 2023b; Lipman et al., 2022; Esser et al., 2024) and diffusion models (Song & Ermon, 2019; Ho et al., 2020) offer a powerful framework for learning complex data distributions $q$. The core idea of flow matching (Lipman et al., 2022) is to construct a continuous sequence from $\mathbf{x}_0 \sim \mathcal{N}(0, 1)$ to $\mathbf{x}_1 \sim q$ by linear interpolation: $\mathbf{x}_t = t\mathbf{x}_1 + (1 - t)\mathbf{x}_0$. The velocity field can be represented as $\mathbf{u}_t = \mathbf{x}_1 - \mathbf{x}_0$. A network is trained to learn the time-dependent velocity $\mathbf{v}(\mathbf{x}_t, t)$) by minimizing

$$\mathrm{E}[\|\mathbf{v}(\mathbf{x}_t, t)) - (\mathbf{x}_1 - \mathbf{x}_0)\|_2^2]. \tag{1}$$

The denoising process transforms a sample from a standard Gaussian distribution into a clean data sample progressively. By numerically integrating the learned velocity $\mathbf{v}(\mathbf{x}_t, t)$), from timestep $0, \cdots, t_1, \cdots, t_2$ to endpoint 1, we can get $\mathbf{x}_{t_1}, \mathbf{x}_{t_2}$ and $\mathbf{x}_1$ according to $\frac{d\mathbf{x}_t}{dt} = \mathbf{v}(\mathbf{x}_t, t)$.

**DiT patchification.** The diffusion transformer (Peebles & Xie, 2023) for estimating $\mathbf{v}(\mathbf{x}_t, t)$) consists of three main components: $\mathrm{Patchify} \rightarrow \mathrm{DiT\ blocks} \rightarrow \mathrm{Unpatchify}$.

$\mathrm{Patchify}$ is a process of converting the spatial input, noisy latents in diffusion, into $L$ tokens. Each token is represented by a vector of dimension $d$, obtained by linearly projecting each patch representation of dimension $p \times p \times C$. The patch size $p \times p$ determines the number of tokens: $L = (I/p)^2$. $I \times I$ is the spatial size of the input noisy latent. The computation complexity of DiT depends on the number of tokens $L$, and thus the patch size $p \times p$. DiT (Peebles & Xie, 2023) selects the patch size $2 \times 2$ for good balance between performance and efficiency. $\mathrm{Unpatchify}$ is a reverse process converting the $d$-dimensional representation output from DiT blocks back to the noisy latent space, e.g. $p \times p \times C$ for velocity estimation. Figure 3 illustrates the $\mathrm{Patchify}$ operation.

## 3.2 PYRAMIDAL PATCHIFICATION FLOW

**Pyramidal patchification.** Our approach, Pyramidal Patchification Flow (PPFlow), divides the timesteps into multiple stages. We use a three-stage example,$\{[0, t_{s_1}), [t_{s_1}, t_{s_2}), [t_{s_2}, 1]\}$, for describing our approach. We illustrate our approach in Figure 2 (a). We adopt a three-level pyramid way to form patch sizes for each stage: large, medium, small patch sizes, $p_{s_1} \times p_{s_1}$, $p_{s_2} \times p_{s_2}$, and $p_{s_3} \times p_{s_3}$ (set to be $2 \times 2$ as the normal DiT in our implementation), for the three stages.

Each patch is a representation vector of dimension $d_{s_i} = Cp_{s_i}^2$. We keep the dimensions of the token representations $d$ the same for all the three stages. The linear projection matrices, mapping patch representations to token representations, are of different sizes for the three stages: $\mathbf{W}_{s_1} \in \mathbb{R}^{d \times d_{s_1}}$, $\mathbf{W}_{s_2} \in \mathbb{R}^{d \times d_{s_2}}$, and $\mathbf{W}_{s_3} \in \mathbb{R}^{d \times d_{s_3}}$. Each stage has its own projection matrices. Similarly, each stage has its own linear projection matrices for $\mathrm{Unpatchify}$.

**Complexity.** The patch size change does not affect the token representation dimension, the structures and parameters of DiT blocks. In our approach, all the parameters in the DiT blocks are shared for all the three stages. In summary, our approach adopts different linear projections in $\mathrm{Patchify}$ and $\mathrm{Unpatchify}$, and the same parameters for DiT blocks in the three stages.

The costs of linear projections in $\mathrm{Patchify}$ (and $\mathrm{Unpatchify}$) are the same for the three stages:

$$L_s \times d_s \times d = (I/p_s)^2 \times (p_s^2 \times C) \times d = I^2 C d. \tag{2}$$

This indicates that the costs of linear projections do not depend on the patch size. Differently, the cost of DiT blocks is dependent on the number of tokens: the time complexity of linear projections and MLPs is linear with respect to the number of tokens, and the complexity of self-attention is

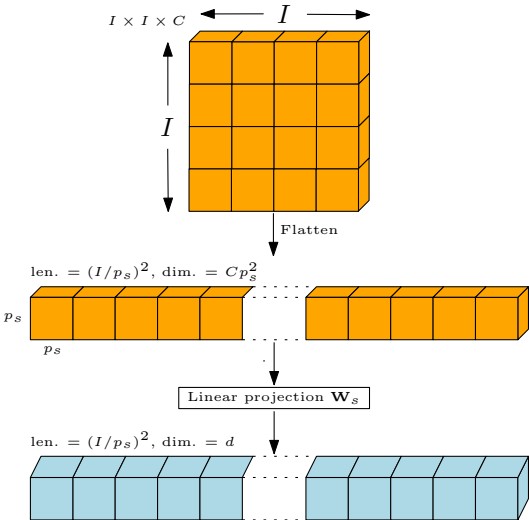

Figure 3: Patchify. After flattening a noisy latent, the layer maps the $p_s \times p_s$ patch representation into a $d$-dimensional token representation through a linear projection $\mathbf{W}_s \in \mathbb{R}^{d \times d_s}$ ($d_s = Cp_s^2$). Unpatchify is a reverse process, mapping the token representation, output from DiT blocks to the predictions. For example, for the velocity predictions, the linear projection matrix is of size $d_s \times d$: $\mathbf{W}_s^u \in \mathbb{R}^{d_s \times d}$.

quadratic. The complexity of each block is:

$$\mathcal{O}(L_s^2 d + L_s d). \tag{3}$$

This indicates that reducing the number of tokens with larger patch sizes effectively reduces the computation complexity. The DiT-XL/2 model (with feature dimension $d = 1152$ and $n = 28$ layers), approximately 99.8% of the computational FLOPs lie in the DiT blocks.

In our approach, the number of tokens at the high-noise stage is smaller. The number of tokens at the low-noise stage is larger and is the same as normal DiTs. Thus, the time complexity of our approach is much lower than normal DiTs for both training and testing. The two-level and three-level PPFlow reduce the FLOPs by 37.8% and 50.6% for $256 \times 256$ image generation.

### 3.3 TRAINING AND INFERENCE

We implement two training regimes and recommend initializing from a pretrained DiT when available, as this preserves performance while only needs limited training cost. In all cases, the flow-matching objective and the stage schedule (time partitions and patch sizes) remain fixed during training.

**Training from scratch.** We train the linear projections in Patchify/Unpatchify and the parameters in the shared DiT blocks by initializing the parameters as normal DiT training and using the standard training setting. The linear projections in Patchify/Unpatchify for different patch sizes are trained only using the noisy latents of the timesteps corresponding to the patch size.

**Training from pretrained DiT.** We copy the weights in DiT blocks from normal DiTs to our approach. The linear projections in Patchify are initialized by averaging. Suppose the patch size is $2 \times 2$ in normal DiTs, and the patch size in the second stage in our approach is $p_{s_2} \times p_{s_2} = 4 \times 4$. The linear projection matrix in the second stage is initialized as: $\mathbf{W}_2 = \frac{1}{4}[\mathbf{W}, \mathbf{W}, \mathbf{W}, \mathbf{W}]$. Here, $\frac{1}{4}$ comes from $\frac{2 \times 2}{4 \times 4}$, This intuitively means that four $2 \times 2$ patches are averaged and then mapped to the $d$-dimensional token.

The linear projection matrix in Unpatchify is initialized by duplicating: $\mathbf{W}_2^u = [(\mathbf{W}^u)^\top, (\mathbf{W}^u)^\top, (\mathbf{W}^u)^\top, (\mathbf{W}^u)^\top]^\top$, where $\mathbf{W}^u \in \mathbb{R}^{d_s \times d}$. This intuitively means that the outputs for the four $2 \times 2$ patches are initially the same. Such initializations are easily extended to other patch sizes.

**Inference.** The sampling process is almost the same as the normal DiTs: start from a noise $\mathbf{x}_0$, e.g., Gaussian noise, and gradually generate less noisy samples $\mathbf{x}_{t_1}, \mathbf{x}_{t_2}, \ldots, \mathbf{x}_{t_K}$ ($t_K = 1$) from timesteps $t_1, t_2, \ldots, t_K$, till getting a clear sample $\mathbf{x}_1$. The only difference lies in that the Patchify and Unpatchify operations for different stages are different: use the patch size and the linear projections that correspond to the timesteps for sampling.

Pyramidal Flow and PixelFlow (Chen et al., 2025c; Jin et al., 2024) need a carefully-designed renoising trick to handle the representation size change issue across stages. Our approach keeps the spatial size of the latent representations unchanged and thus does not need such a renoising scheme.

## 4 EXPERIMENTS

### 4.1 EXPERIMENTAL SETUP

**Datasets.** We train class-conditional PPFlow models on ImageNet (Deng et al., 2009). Following latent diffusion practice, we use the Stable Diffusion VAE encoder (Rombach et al., 2022) to map an RGB image $x \in \mathbb{R}^{H \times W \times 3}$ to a latent $z \in \mathbb{R}^{H/8 \times W/8 \times 4}$, with $H \in \{256, 512\}$. For text-to-image, we use curated LAION (Schuhmann et al., 2022), JourneyDB (Sun et al., 2023), BLIP3o-60k (Chen et al., 2025b) and 1M images generated by FLUX.1 dev (Labs, 2024) utilizing prompts from LLaVA-pretrain (Liu et al., 2023a).

**Implementation.** We adopt SiT (Ma et al., 2024) as the main baseline and starting point for PPFlow. For class-conditional models, we evaluate at $256 \times 256$ and $512 \times 512$. Unless otherwise noted, models are trained with AdamW (Kingma & Ba, 2014; Loshchilov & Hutter, 2017), learning rate $1 \times 10^{-4}$, no weight decay, batch size 256, and horizontal flips as the only augmentation. We maintain EMA with decay 0.9999 and report all metrics with EMA weights.

Models are denoted by capacity and stage count, e.g., PPF-XL-2 is an XLarge model with two patchification stages. For two-level, $4 \times 4$ patches for $t \in [0, 0.5)$; $2 \times 2$ patches for $t \in [0.5, 1.0]$. For three-level, $4 \times 4$ for $t \in [0, 0.5)$; $4 \times 2$ for $t \in [0.5, 0.75]$; $2 \times 2$ for $t \in (0.75, 1.0]$. We use a stage-wise CFG schedule and Patch n' Pack (Dehghani et al., 2023) to pack variable-length token sequences into batches, reducing the training FLOPs in an iteration compared with normal DiT.

For text-to-image, we integrate two-stage PPFlow into pretrained FLUX.1-dev model. We adopt a progressive training scheme, where the model is trained sequentially at resolutions of $512 \times 512, 1024 \times 1024, 2048 \times 2048$, with corresponding batch sizes of 256, 128, and $16 \times 4$ (4 gradient accumulation steps), respectively. We use a fixed learning rate of $1 \times 10^{-5}$ for the entire 90k training iterations.

**Metrics.** For class-conditional generation, we report FID-50K (Heusel et al., 2017), IS (Salimans et al., 2016), sFID (Nash et al., 2021), and Precision/Recall (Kynkäänniemi et al., 2019) using ADM's TensorFlow evaluation suite (Dhariwal & Nichol, 2021). For text-to-image, we report GenEval (Ghosh et al., 2023), DPG-bench (Hu et al., 2024), and T2I-CompBench (Color/Shape/-Texture).

### 4.2 RESULTS ON CLASS-CONDITIONAL IMAGE GENERATION

We study two training regimes: (i) training from scratch and (ii) training from pretrained DiTs (SiT-B/2, SiT-XL/2, DiT-XL/2).

**Training from Scratch.** We train PPF-B-2 and PPF-B-3 for 11M steps from scratch, the PPF-B models achieve better or comparable FID-50K to SiT-B/2 at substantially lower testing FLOPs, while consistently improving IS. Quantitative results are summarized in Table 1. We use stage-wise CFG schedules: $[1.5, 3.5]$ for PPF-B-2 and $[1.5, 3.5, 4.0]$ for PPF-B-3, inspired by adaptive guidance (Chang et al., 2022; Kynkäänniemi et al., 2024; Wang et al., 2024). Visual comparisons with the same noise inputs (Appendix Figure 5) show that pyramidal patchification preserves image quality while improving compute efficiency. Overall, PPFlow attains approximately $1.6\times$ (two-level) and $2.0\times$ (three-level) inference speedups relative to SiT-B/2 while keeps comparable generation quality.

| Method | Training steps | Training FLOPs (%) | Testing FLOPs (%) | FID-50k ↓ | sFID ↓ | IS ↑ | Pre. ↑ | Rec. ↑ |
|---|---|---|---|---|---|---|---|---|
| SiT-B/2 | 7M | 100 | 100 | 4.46 | **4.87** | 180.95 | 0.78 | **0.57** |
| PPF-B-2 | 7M | 62.5 | 62.0 | 4.12 | 5.71 | 211.36 | 0.79 | 0.53 |
| PPF-B-2 | 11M | 98.2 | 62.0 | **3.83** | 5.70 | 223.00 | 0.81 | 0.53 |
| PPF-B-3 | 7M | 50.0 | 49.1 | 4.71 | 5.78 | 212.84 | 0.81 | 0.49 |
| PPF-B-3 | 11M | 78.5 | 49.1 | 4.43 | 5.85 | **230.72** | **0.83** | 0.48 |

Table 1: Train from scratch comparison of our approach to normal SiT-B/2. The result of SiT-B/2 is from (Ma et al., 2024; Dao et al., 2023). Our approach, trained from scratch, with more training steps but smaller training FLOPs, performs similarly: better for three metrics and worse for other two metrics. Our approach obtains $1.6\times$ and $2.0\times$ inference speedup.

| Method | Size. | Training steps | Training FLOPs (%) | Testing FLOPs (%) | FID-50k ↓ | sFID ↓ | IS ↑ | Pre. ↑ | Rec. ↑ |
|---|---|---|---|---|---|---|---|---|---|
| SiT-B/2 | 256 | - | 100 | 100 | 4.46 | **4.87** | 180.95 | 0.78 | **0.57** |
| PPF-B-2 | 256 | 1M | 8.9 | 62.0 | **4.22** | 5.49 | **252.10** | **0.85** | 0.49 |
| PPF-B-3 | 256 | 1M | 7.1 | 49.1 | 4.57 | 6.06 | 236.53 | 0.83 | 0.48 |
| SiT-XL/2 | 256 | - | 100 | 100 | 2.15 | **4.60** | 258.09 | **0.81** | 0.60 |
| PPF-XL-2 | 256 | 1M | 8.9 | 62.6 | **1.99** | 5.52 | 271.62 | 0.78 | 0.63 |
| PPF-XL-3 | 256 | 1M | 7.1 | 49.4 | 2.23 | 5.50 | **286.67** | 0.78 | **0.64** |
| DiT-XL/2 | 512 | - | 100 | 100 | 3.04 | **5.02** | 240.82 | 0.84 | 0.54 |
| PPF-XL-2 | 512 | 400k | 7.6 | 58.7 | **3.01** | 5.24 | **249.98** | 0.84 | 0.54 |
| PPF-XL-3 | 512 | 400k | 5.8 | 45.4 | 3.06 | 5.31 | 249.91 | 0.83 | 0.53 |

Table 2: Train from pretrained model comparison of our approach to normal SiTs and DiT. Our approach is trained from the corresponding pretrained model with less than $10\%$ training FLOPs of the pretrained model (7M training steps for 256 size, 3M training steps for 512 size). Our approach achieves overall comparable performance with less testing FLOPs.

**Training from Pretrained DiTs.** We train PPFlow from pretrained SiTs with only $\leq 10\%$ of the pretraining FLOPs and evaluate at testing reduced FLOPs (Table 2). For multiple model configs (B and XL), multiple image resolution (256 and 512), PPFlow can save 37.4% - 41.3% testing FLOPs for two-level and 50.6% - 54.6% for three-level with comparable FID score and better IS.

Stage-wise CFG schedules used here are $[1.0, 3.0]$ for PPF-XL-2 and $[1.0, 3.5, 3.75]$ for PPF-XL-3. Visual comparisons can be seen in Appendix (Figure 6) confirm quality is preserved with substantially lower compute.

## 4.3 RESULTS ON TEXT-TO-IMAGE

We integrate two-stage PPFlow into the pretrained FLUX.1-dev model at multiple resolutions: $512 \times 512, 1024 \times 1024, 2048 \times 2048$. We fine-tune the FLUX.1 model using our collected dataset as the baseline of our approach, and we denote the model as FLUX.1-ft. We train PPFlow from the model FLUX.1-ft.

From Table 3, one can see that PPFlow's results (GenEval for compositional aspects, DPG Bench for complex textual prompts and T2I-CompBench for alignment with complex semantic relationships) are comparable with the FLUX.1-ft, consistent to the observation in class-conditional image generation. The results on three different resolutions indicate that PPFlow is scalable to higher-resolution generation, and the testing FLOPs progressively decrease as the input resolution is increasing.

## 4.4 ABLATION STUDY

**Patch-Level embedding and stage-wise CFG.** We ablate two network components on PPF-B-2 training for 1M steps from pretrained SiT-B/2 (Table 6). Adding a learned patch-level (stage) embedding improves FID from 4.44 to 4.30. Adding stage-wise CFG further improves FID to 4.22 and markedly boosts IS.

| Method | Testing FLOPs (%) | GenEval ↑ | DPG Bench ↑ | T2I-CompBench | | |
|---|---|---|---|---|---|---|
| | | | | Color ↑ | Shape ↑ | Texture ↑ |
| *512 resolution* | | | | | | |
| FLUX.1 | 100 | 0.67 | 82.51 | 0.7534 | 0.5060 | 0.6306 |
| FLUX.1-ft | 100 | 0.68 | 82.87 | 0.7556 | **0.5070** | **0.6310** |
| PPF-FLUX.1 | 62.2 | **0.68** | **82.90** | **0.7560** | 0.5066 | **0.6310** |
| *1024 resolution* | | | | | | |
| FLUX.1 | 100 | **0.68** | 83.14 | 0.7529 | 0.5056 | 0.6312 |
| FLUX.1-ft | 100 | **0.68** | 83.89 | **0.7568** | 0.5098 | 0.6400 |
| PPF-FLUX.1 | 59.1 | **0.68** | **84.00** | 0.7566 | **0.5100** | **0.6423** |
| *2048 resolution* | | | | | | |
| FLUX.1 | 100 | 0.66 | 82.75 | 0.7510 | 0.5066 | 0.6300 |
| FLUX.1-ft | 100 | **0.67** | 83.02 | 0.7515 | 0.5082 | 0.6308 |
| PPF-FLUX.1 | 53.9 | **0.67** | **83.10** | **0.7520** | **0.5086** | **0.6310** |

Table 3: Performance of PPFlow applied to FLUX.1-dev across multiple resolutions ($512 \times 512$ to $2048 \times 2048$). We compare against FLUX.1-ft, a fine-tuned baseline in a same dataset, on three benchmarks: GenEval, DPG Bench, and T2I-CompBench. The results indicate that PPFlow's performance is comparable with the fine-tuned model, showing its potential for text-to-image task.

| Method | Testing FLOPs (%) | FID-50k ↓ |
|---|---|---|
| DiT-XL/2 | 100 | 2.27 |
| FlexiDiT | 64 | 2.25 |
| FlexiDiT | 46 | 2.64 |
| PPF-DiT-XL/2 | 62.6 | **2.15** |
| PPF-DiT-XL/3 | 49.4 | 2.31 |

Table 4: Comparison of our approach to FlexiDiT based on DiT-XL/2 of 256 resolution. At around 63% FLOPs budget, PPFlow achieves an FID of 2.15, outperforming Flex-iDiT's 2.25. Further, at around 50% FLOPs, PPFlow's FID of 2.31 is substantially better than 2.64 FID of FlexiDiT.

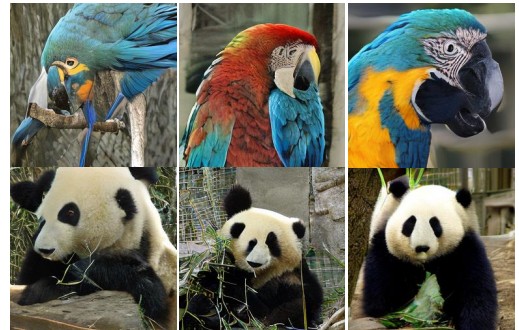

Figure 4: Visualization comparison between pyramid representations + renoising (left), Lumina-Video method (middle) and our PPF-B-2 (right). One can see that right generation results are visually better.

**Number of patch sizes.** We explore the influence of increasing number of stages on PPFlow, specifically, varying stage number from 2 to 5 on top of pretrained SiT-B/2 (Table 7). Increasing stages consistently lowers test-time FLOPs. With additional training, models recover the two-stage quality at significantly reduced FLOPs, demonstrating a trade-off by stage count and training budget.

**Timestep segmentation.** We investigate different time segmentations for two- and three-stage PPFlow (Table 8). The results show that for PPFlows with different time segmentations, the testing FLOPs vary, and they need different training steps to maintain comparable performance as the normal SiT-B. In general, lower FLOPs needs more training steps.

**Pyramid representations.** We implement PyramidalFlow (Jin et al., 2024) on two-stage variants of the SiT-B and train them from scratch for 1M steps. The results in Table 5 show: PPFlow attains the best FID, sFID, IS, and Precision/Recall. Only using pyramid representations produces blurred images suffering from block-like artifacts (Jin et al., 2024). The test-time renoising trick only partially alleviates the issue.

| Method | Training steps | FID-50k ↓ | sFID ↓ | IS ↑ | Pre.↑ | Rec.↑ |
|---|---|---|---|---|---|---|
| Pyramid Rep. | 1M | 164.48 | 61.58 | 8.29 | 0.09 | 0.14 |
| Pyramid Rep. + Renoising | 1M | 27.69 | 11.16 | 73.20 | 0.55 | 0.57 |
| Lumina-Video method | 1M | 18.77 | 6.17 | 79.12 | 0.64 | 0.57 |
| PPF-B-2 | 1M | **15.68** | **5.82** | **88.78** | **0.65** | **0.58** |

Table 5: Comparison of our approach to pyramid representation-based flow, Lumina-Video. The results at 1M training steps from scratch are reported. Overall performance of PPFlow is better.

|  | Training steps | FID-50k ↓ | sFID ↓ | IS ↑ | Pre. ↑ | Rec. ↑ |
|---|---|---|---|---|---|---|
| PPF-B-2 | 1M | 4.44 | 5.55 | 201.12 | 0.80 | 0.55 |
| + level Emb. | 1M | 4.30 | 5.50 | 212.70 | 0.81 | 0.55 |
| + stage CFG | 1M | 4.22 | 5.49 | 252.10 | 0.85 | 0.49 |

Table 6: Ablation study for patch-level embedding and stage-wise CFG. Level embedding and stage-wise CFG can help improve FID and IS in PPFlow.

| Model | Training steps | Testing FLOPs (%) | FID-50k ↓ | sFID ↓ | IS ↑ | Pre. ↑ | Rec. ↑ |
|---|---|---|---|---|---|---|---|
| SiT-B/2 | - | 100 | 4.46 | 4.87 | 180.95 | 0.78 | 0.57 |
| PPF-B-2 | **1M** | 62.0 | 4.22 | 5.49 | 252.10 | 0.85 | 0.49 |
| PPF-B-3 | **1M** | 49.1 | 4.57 | 6.06 | 236.53 | 0.83 | 0.48 |
| PPF-B-4 | 1M | 46.5 | 4.78 | 6.23 | 227.12 | 0.82 | 0.48 |
|  | **2.5M** | 46.5 | 4.41 | 5.77 | 245.12 | 0.83 | 0.49 |
| PPF-B-5 | 1M | 38.3 | 5.30 | 6.66 | 220.13 | 0.80 | 0.49 |
|  | **4M** | 38.3 | 4.51 | 5.99 | 236.89 | 0.83 | 0.48 |

Table 7: Ablation study for different stages in PPFlow. PPFlow can be applied with more stages for more efficient inference with the performance maintained through more training steps.

| Model | Time segmentation | Training steps | Testing FLOPs (%) | FID-50k ↓ | sFID ↓ | IS ↑ | Pre. ↑ | Rec. ↑ |
|---|---|---|---|---|---|---|---|---|
| SiT-B/2 | - | - | 100 | 4.46 | 4.87 | 180.95 | 0.78 | 0.57 |
| PPF-B-2 | [0.50, 1.0] | 1.0M | 62.0 | 4.22 | 5.49 | 252.10 | 0.85 | 0.49 |
|  | [0.25, 1.0] | 0.5M | 80.1 | 4.25 | 5.29 | 249.12 | 0.84 | 0.49 |
|  | [0.75, 1.0] | 2.5M | 43.4 | 4.40 | 5.70 | 242.10 | 0.84 | 0.50 |
| PPF-B-3 | [0.50, 0.75, 1.0] | 1.0M | 49.1 | 4.57 | 6.06 | 236.53 | 0.83 | 0.48 |
|  | [0.25, 0.50, 1.0] | 0.8M | 68.3 | 4.21 | 5.30 | 240.11 | 0.84 | 0.49 |
|  | [0.50, 0.90, 1.0] | 3.0M | 41.8 | 4.59 | 6.02 | 235.57 | 0.81 | 0.50 |

Table 8: Ablation study for time segmentation in PPFlow. The results show PPFlows with different time segmentations, the testing FLOPs vary, and they need different training steps to maintain comparable performance as the normal SiT-B.

**Training with varying patch sizes.** We study the performance of the alternative methods of training with varying patch sizes (Liu et al., 2025; Anagnostidis et al., 2025), which train the model of each patch size for all the timesteps. We implement the training process in Lumina-Video (Liu et al., 2025) whose inference is similar to ours. It trains the models of different patch sizes over all the timesteps, and adopts a shifted-sampling scheme: sample more lower-noise timesteps for training the model with a smaller patch size and less higher-noise timesteps for training the model with a larger patch size. This leads to inconsistency between the training and testing. Our approach, training the model of each patch size for the range of the corresponding timesteps, introduces an additional benefit: the model of a patch size can better handle the corresponding specific noise degrees. From the results in Table 5 and Figure 4, one can see that our approach achieves better results at the 1M training iteration.

The results from FlexiDiT (Anagnostidis et al., 2025) whose training is similar to Lumina-Video (Liu et al., 2025) but without adopting the shift-sampling scheme are reported in Table 4[1]. As FlexiDiT (Anagnostidis et al., 2025) does not report the result from SiT, we apply our method to DiT. Our approach demonstrates superior performance over FlexiDiT at comparable computational levels. At around 63% FLOPs budget, PPFlow achieves an FID of 2.15, outperforming FlexiDiT's 2.25. Further, at around 50% FLOPs, PPFlow's FID of 2.31 is substantially better than the 2.64 FID of FlexiDiT.

**sFID discussion.** In Table. 1 and Table. 2, our sFID score is slightly worse than the baseline. We suppose this is related to the spatial sensitivity that sFID evaluates. We analyze this from three

---

[1]FlexiDiT is not open-sourced. The results are from the paper (Anagnostidis et al., 2025).

observations: (1) Continue the training to see how sFID changes with more training. After 11M training steps from scratch, the FID score stabilizes, while the sFID score progressively reduces to 5.03, approaching the 4.87 baseline of the normal SiT-B/2 model. (2) From visualization results, no obvious implications except some slight spatial difference in the position and size of the main object compared to the baseline shown in Fig. 5 and Fig. 6; (3) The *position* evaluation metric (0.20) in GenEval for PPF-FLUX.1 is same with FLUX.1-ft and FLUX.1. This gives a more evidence: in the SoTA text-to-image model, our approach does not influence the image generation quality in position relationships.

## 5 CONCLUSION

We introduce PPFlow, a pyramidal patchification scheme that adaptively reduces token counts at high-noise timesteps while maintaining latent representation resolutions unchanged across timesteps. PPFlow keeps inference identical to standard DiT — avoiding re-noising and resolution jumps. Across training from scratch and from pretrained models, PPFlow achieves 1.6–2.0 denoising speedups with comparable or improved image quality for class-conditional image generation. Our approach is also demonstrated for text-to-image generation: nearly 50% sampling cost reduction and image generation quality kept. The approach is simple and easily-implemented.

## 6 ACKNOWLEDGMENTS

This work was supported in part by the Shanghai Municipal Commission of Economy and Informatization (No. 2025-GZL-RGZN-BTBX-01011).

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

## A   USAGE OF LLM

During the preparation of this manuscript, we utilized Large Language Models (LLMs) as a writing assistant. The tool was employed to enhance grammar, clarity, and overall readability. The authors reviewed and edited all AI-generated suggestions to ensure that the final text accurately and faithfully reflects our original ideas and contributions.

## B   LIMITATIONS AND FUTURE WORK

Our evaluation is limited to class-conditional image generation and text-to-image generation, and a fixed, manually designed stage schedule with discrete patch sizes; generalization to video and other visual domains as well as to alternative noise/solver settings, remains untested. Future work includes: (i) jointly learning stage boundaries and patch sizes end-to-end, (ii) introducing stage-aware conditioning or selectively untying parameters across stages, and (iii) integrating PPFlow with step-reduction, distillation, and compression techniques to further improve the efficiency–quality trade-off.

## C   VISUALIZATION RESULTS

Figure 5 and Figure 6 report the results for training from scratch and training from pretrained models of our approach as well as the results for the normally-trained SiT models.

Increasing patchification stages

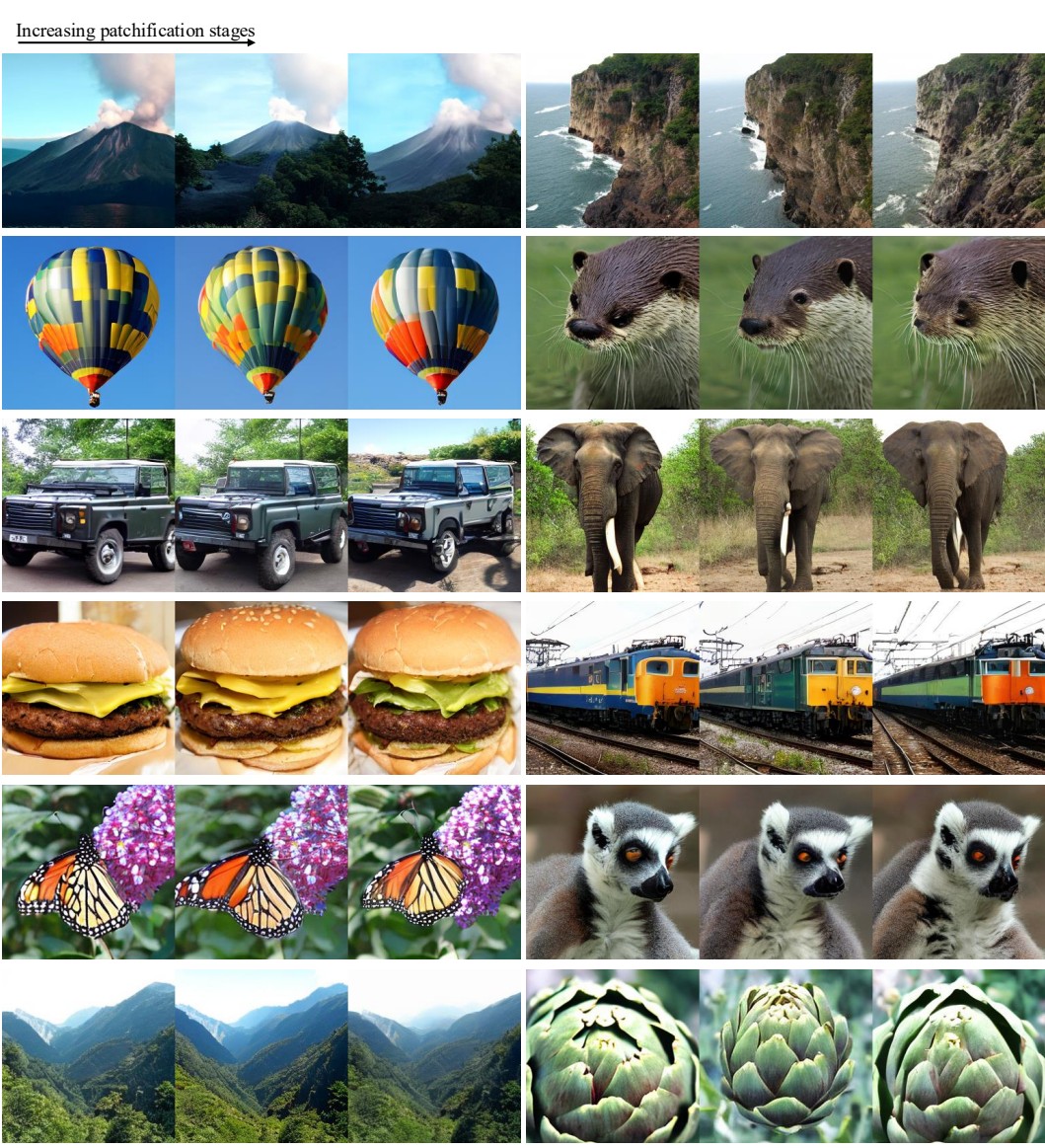

Figure 5: Visualization results for normal SiT-B/2, PPF-B-2, and PPF-B-3. The results are sampled from the same noise. The models of our approach are trained from scratch. The results of the three methods are visually comparable.

Increasing patchification stages

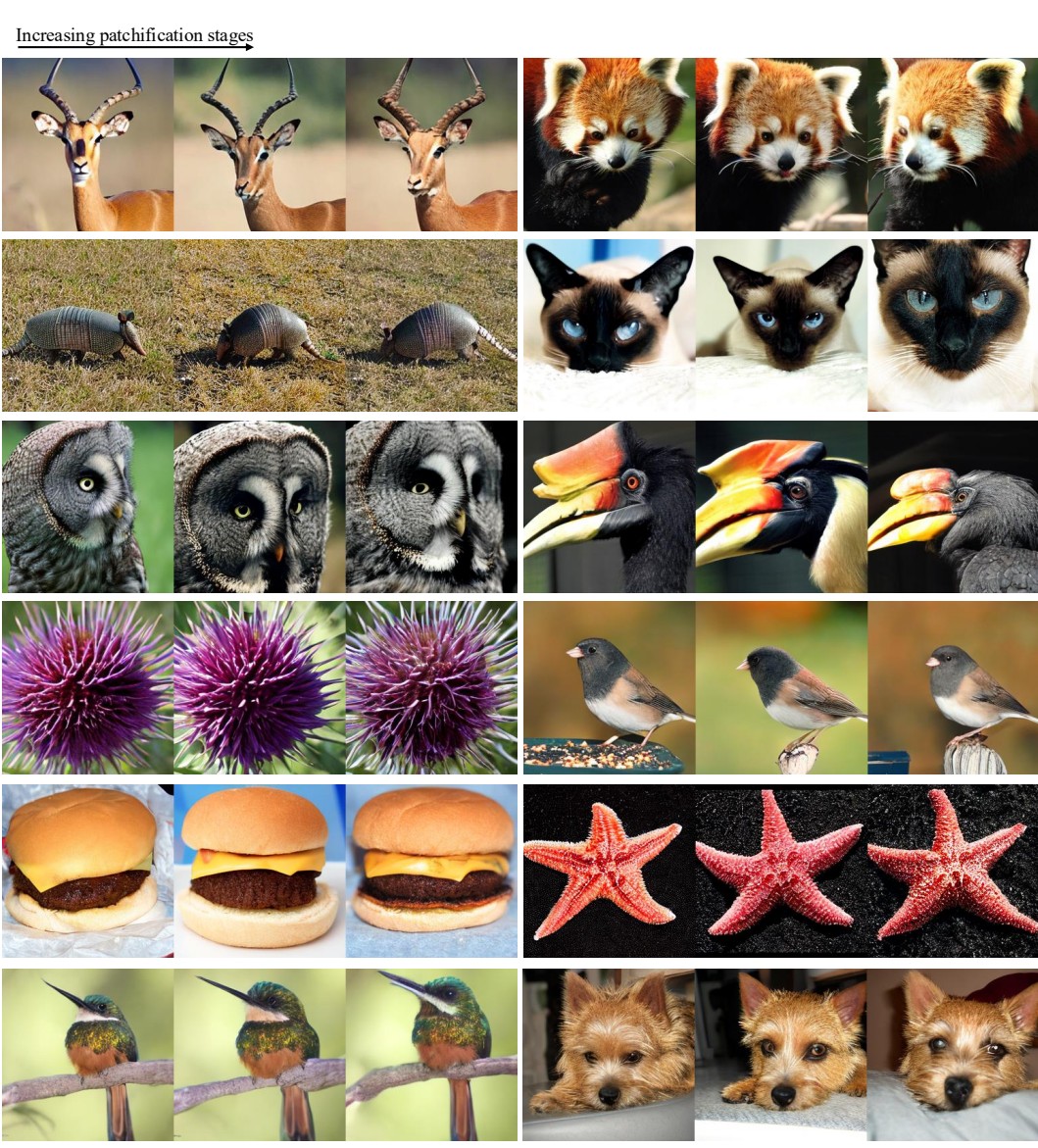

Figure 6: Visualization results for normal SiT-XL/2, PPF-XL-2, and PPF-XL-3. The results are sampled from the same noise. Our models are trained from pretrained normal SiT-XL/2. The results of the three methods are visually comparable.

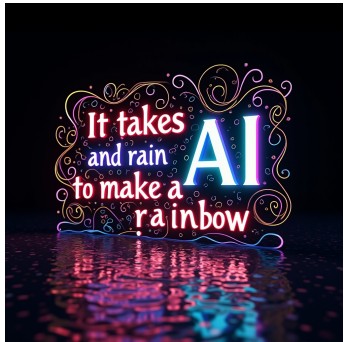 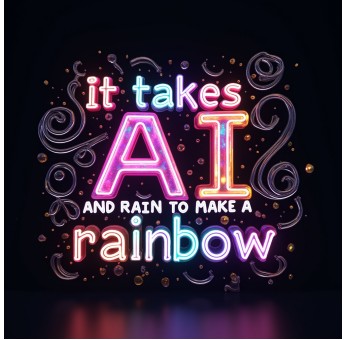 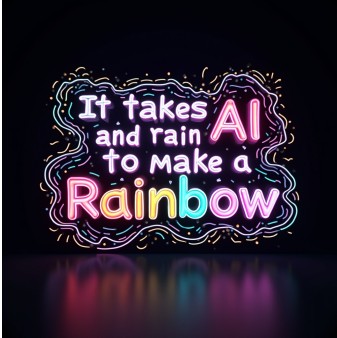

(a) A visually striking digital art piece featuring the phrase "It takes AI and rain to make a rainbow" set against a deep black background...

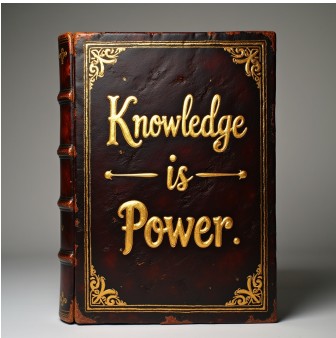 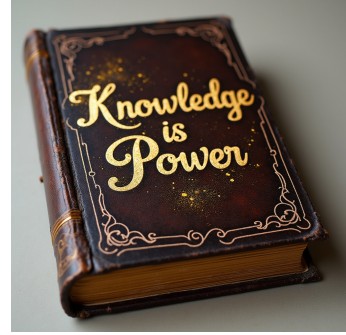 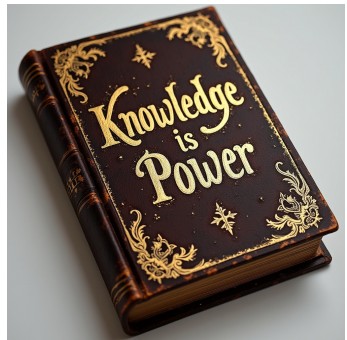

(b) A beautifully aged antique book is positioned carefully for a studio close-up, revealing a rich, dark brown leather cover. The words "Knowledge is Power" are prominently featured in the center with thick...

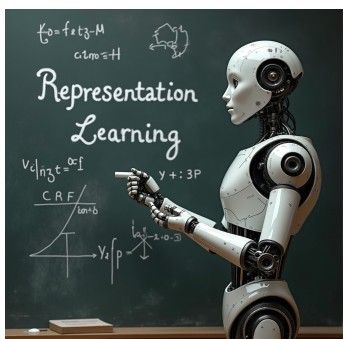 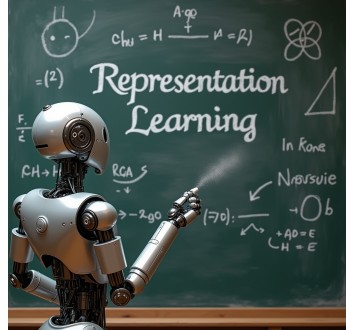 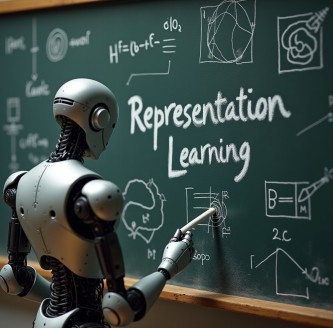

(c) In the image, a sleek, metallic humanoid robot stands before a dusty chalkboard, on which it has carefully scrawled 'Representation Learning' in eloquent cursive...

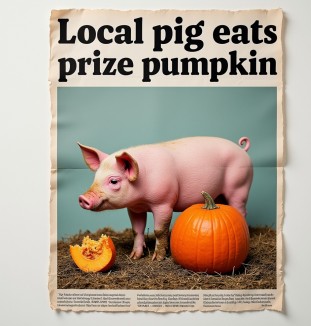 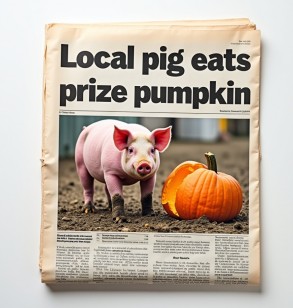 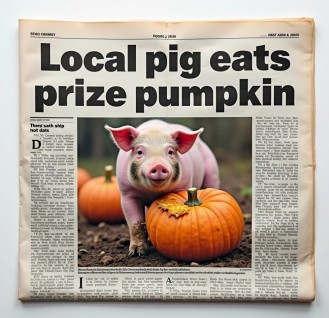

(d) An image of a newspaper lies flat, its bold headline 'Local pig eats prize pumpkin' emblazoned across the top in large lettering....

Figure 7: Example results demonstrate that PPFlow maintains visual quality, specifically for text rendering. Left: FLUX.1; Middle: FLUX.1-ft; and Right: PPF-FLUX.1.

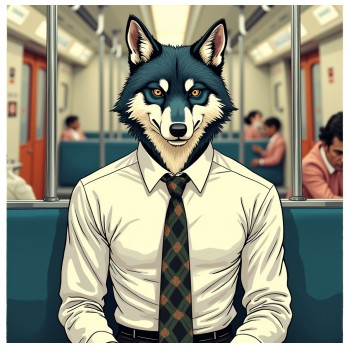 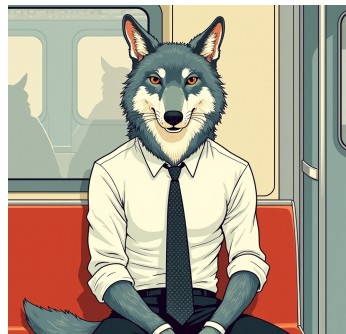 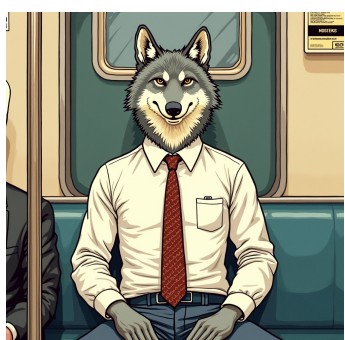

(a) An anthropomorphic wolf, clad in a crisp white shirt and a patterned tie...

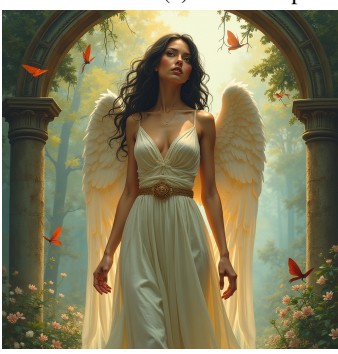 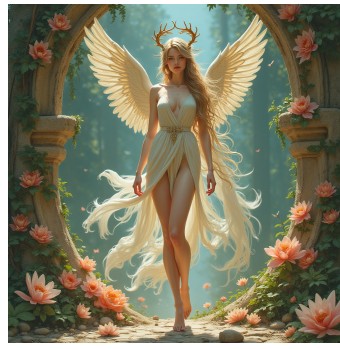 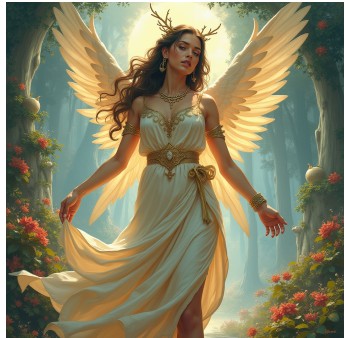

(b) An imaginative digital artwork that features a fantasy female character, stylized with the intricate detail and ethereal qualities reminiscent of Peter Mohrbacher's angelic designs...

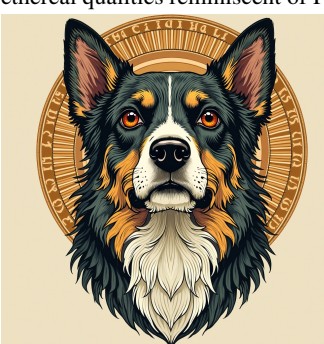 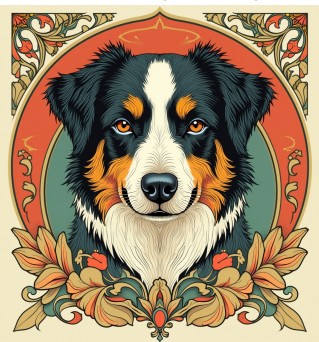 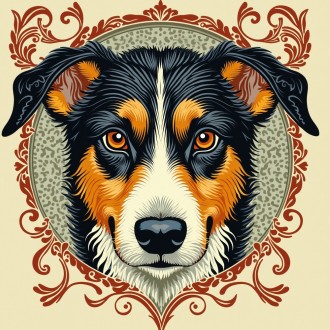

(c) A meticulously crafted Art Nouveau screenprint featuring a dog's face, characterized by its remarkable symmetry and elaborate detailing...

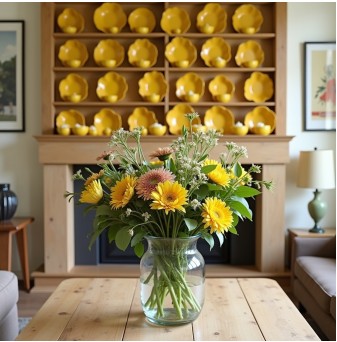 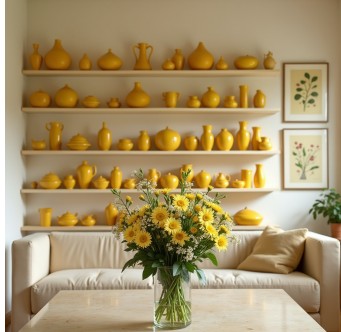 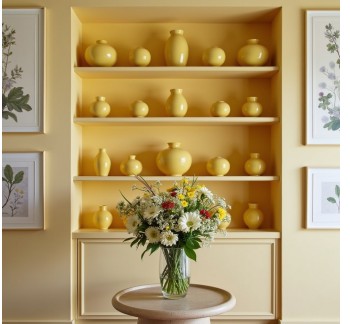

(d) An interior space featuring a collection of yellow ceramic ornaments neatly arranged on a shelf. In the center of the room stands a table with a clear glass vase filled with a bouquet of fresh flowers...

Figure 8: Example results demonstrate that PPFlow maintains visual quality. Left: FLUX.1; Middle: FLUX.1-ft; and Right: PPF-FLUX.1.

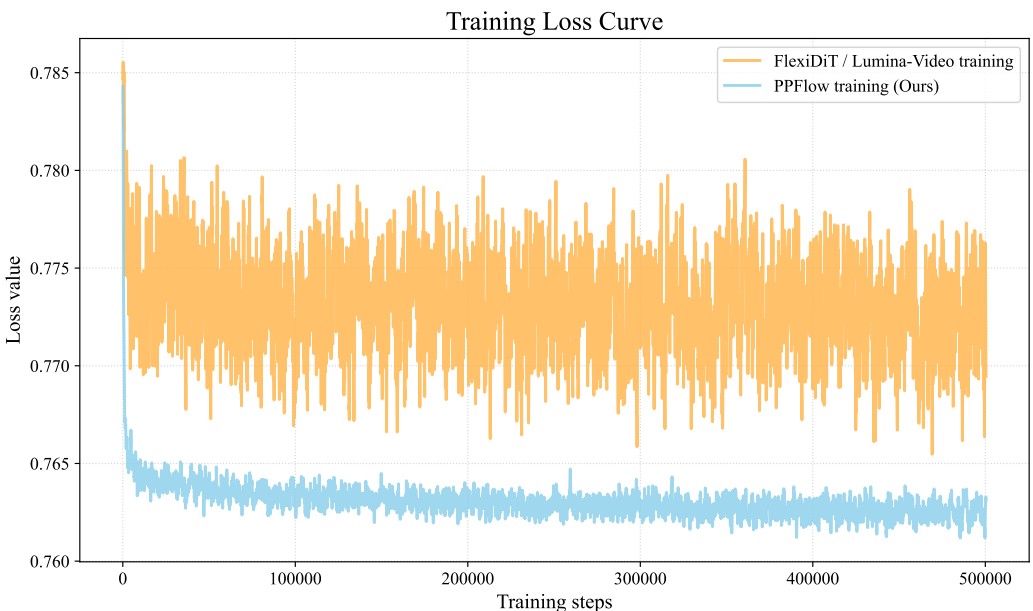

Figure 9: Training loss comparison. PPFlow (blue) achieves consistently lower loss and stability than FlexiDiT (orange). This validates that our training strategy works better, avoiding the conflicting optimazation in handling large and small patch sizes simultaneously.

```python
# Extract baseline model patchification projection weights
source_proj_weight = state_dict["x_embedder.proj.weight"]
source_proj_bias = state_dict["x_embedder.proj.bias"]

# Initialize patchification projection (deal with larger patch size)
    weights with scaled and repeated weights
initialize_layer_weights(
    source_proj_weight,
    model.x_embedder.proj_large_patch,  # new weight
    repeat_factor=4,
    scale_factor=4.0,
    dim=1
)
# Initialize bias
initialize_layer_bias(
    source_proj_bias,
    model.x_embedder.proj_large_patch
    )

def initialize_layer_weights(source_weight, target_layer, repeat_factor,
                            scale_factor=1.0, dim=1):
    """
    Initializes the weights of a target layer by scaling and repeating.
    """
    with torch.no_grad():
        # Scale and repeat the source weight
        target_weight = source_weight / scale_factor
        initialized_weight = target_weight.repeat_interleave(
    repeat_factor, dim=dim)

        # Copy the initialized weights to the target layer
        target_layer.weight.data.copy_(initialized_weight)

def initialize_layer_bias(source_bias, target_layer, repeat_factor=1, dim
    =0):
    """
    Initializes the bias of a target layer.
    """
    with torch.no_grad():
        if repeat_factor > 1:
            initialized_bias = source_bias.repeat_interleave(
    repeat_factor, dim=dim)
        else:
            initialized_bias = source_bias

        # Copy the initialized bias to the target layer
        target_layer.bias.data.copy_(initialized_bias)
```

Listing 1: PyTorch implementation for initializing the patchification projection layers.

