# OpenReview forum: "Pyramid Patchification Flow for Visual Generation"
_ICLR.cc/2026/Conference — ICLR 2026 Poster_

### Official Review · Reviewer_R4cM · 2025-10-25

**Soundness:** 2
**Presentation:** 3
**Contribution:** 2
**Rating:** 4
**Confidence:** 4

**Summary:**

This paper introduces "Pyramidal Patchification Flow" (PPFlow), a method designed to improve the inference efficiency of Diffusion Transformers (DiT) / Flow Transformers (e.g., SiT, FLUX.1). Standard DiTs use a fixed patch size (e.g., 2x2) across all denoising timesteps, resulting in a constant number of tokens. PPFlow's core idea is simple: use larger patch sizes (e.g., 4x4) for earlier, higher-noise timesteps and smaller patch sizes (e.g., 2x2) for later, lower-noise timesteps, thereby reducing the token count in early stages and accelerating inference.

The implementation is straightforward:

1. Shared DiT Body: The main Transformer blocks (DiT Blocks) are shared across timesteps (and thus, different patch sizes).

2. Independent Patchify/Unpatchify Layers: Separate linear projection layers ($W_s$ for Patchify, $W_s^u$ for Unpatchify) are learned for each distinct patch size (e.g., one set for 4x4, another for 2x2).

A key distinction from methods like Pyramidal Flow (Jin et al., 2024) is that PPFlow operates consistently on full-resolution latent representations, merely changing how they are patchified, whereas Pyramidal Flow operates on pyramid representations of varying resolutions. This allows PPFlow to avoid "jump points" caused by resolution changes and eliminates the need for complex re-noising tricks during sampling.

Experiments show that fine-tuning PPFlow (three-stage) from a pretrained SiT-XL/2 model requires minimal additional training cost (+7.1% FLOPs) while achieving ~2.02x inference speedup and maintaining comparable image generation quality (FID). Similar speedups are observed when training from scratch. When applied to the text-to-image model FLUX.1, PPFlow achieves 1.61-1.86x speedup across various resolutions with comparable quality.

**Strengths:**

1. **Simple and Effective**: The core idea—dynamically adjusting patch size based on noise level to reduce token count—is intuitive, simple, and easy to implement (only modifying Patchify/Unpatchify layers).

2. **Inference Acceleration**: Experimental results (Tables 1, 2, 3) clearly demonstrate that PPFlow achieves substantial inference speedups (approx. 1.6x to 2x) across multiple models (SiT-B, SiT-XL, DiT-XL, FLUX.1) and resolutions (256 to 2048), while maintaining comparable image generation quality (FID). This is a highly practical contribution.

3. **Avoids "Jump Point" Issues**: Compared to methods like Pyramidal Flow that switch between different representation resolutions, PPFlow's consistent operation on full-resolution latents avoids the complexities associated with "jump points" and re-noising sampling strategies, keeping the inference process simple.

4. **Easy Adaptation from Pretrained Models**: The paper shows that PPFlow can be efficiently adapted from pretrained DiT models via fine-tuning (Table 2), requiring only about 7-9% additional training FLOPs to gain significant inference speedups. This greatly increases the practical appeal of the method.

5. **Clear Concept**: The paper is well-written with a clear concept. The experimental design is sound, covering both training-from-scratch and fine-tuning scenarios, validated on class-conditional and text-to-image generation. Ablation studies (Tables 6-8) are reasonably thorough, supporting the design choices.

**Weaknesses:**

1. **Limited Novelty**: While effective, the core idea (adjusting computation/token count based on noise level) is not entirely new. Related works like multi-scale/cascaded Diffusion, Pyramidal Flow (Jin et al., 2024), and especially concurrent works FlexiDiT (Anagnostidis et al., 2025) and Lumina-Video (Liu et al., 2025) explore the exact same core mechanism (time-varying patch sizes). PPFlow's primary novelty lies in its specific implementation (avoiding jump points via full-resolution operation) and its staged training strategy.

2. **Insufficient Comparison with Concurrent Work**: The paper mentions concurrent works FlexiDiT and Lumina-Video in Sec 2 and 4.4. Although it claims PPFlow's training strategy (staged training) is superior to their approach (full-timeline training) and experiments (Tables 4, 5) show better results for PPFlow, the argumentation for why PPFlow's strategy is better, and the trade-offs against the claimed "flexibility" of FlexiDiT, could be discussed more deeply.

3. **Arbitrary Stage Divisions and Patch Sizes**: The timestep points for stage transitions (e.g., 0.5, 0.75) and the patch sizes used (e.g., 4x4, 4x2, 2x2) are manually chosen. The paper lacks sensitivity analysis on these hyperparameters or exploration of more automated selection methods. Increasing the number of stages (Table 7) seems to have diminishing returns and requires more training.

4. **Missing Key Visualizations**: While Table 3 presents metric results for applying PPFlow to FLUX.1 (text-to-image model), the paper lacks corresponding visual samples. Given the subjective nature of text-to-image tasks, adding visual comparisons between PPF-FLUX.1 and the baseline FLUX.1-ft would significantly strengthen the conclusions.

**Questions:**

**Question 1: What are the fundamental trade-offs between PPFlow's staged training and FlexiDiT/Lumina-Video's full-timeline training?**

The paper suggests PPFlow's staged training allows models to better specialize for specific noise levels, and experiments show better performance than a re-implementation of the Lumina-Video approach. However, FlexiDiT claims their full-timeline training offers more "flexibility," potentially allowing the use of any patch size at any timestep during inference. Could the authors elaborate on the trade-offs? Is PPFlow's specialized approach inherently better under fixed schedules, while FlexiDiT's generalist approach offers more adaptability if the inference schedule (patch sizes vs. timesteps) needs modification? A direct experimental comparison of these training strategies (e.g., using PPFlow's architecture but FlexiDiT's training method) would be valuable.

**Question 2: How sensitive are the results to the choice of stage divisions (timesteps) and patch size combinations?**

The current stage divisions and patch sizes are set manually. How much do the final speedup and image quality depend on these specific choices? Are there better combinations? Could these parameters potentially be learned?

**Question 3: How does PPFlow perform on tasks requiring fine spatial structure or long-range dependencies?**

Does using larger patches (which discard early high-frequency information) in high-noise stages negatively impact tasks that require precise spatial layout or fine-grained textures (e.g., complex scenes, generating images with text)? While ImageNet and T2I benchmark results are strong, how does performance hold up in more challenging scenarios? Suggest adding relevant experiments or discussion.

**Question 4: Could 1x1 patch sizes be considered for the final, very low-noise stages?**

To potentially capture the finest details in the last stages (close to t=1), would incorporating a 1x1 patch size (equivalent to operating directly on pixels/tokens) be beneficial? This might further improve image quality but would require careful consideration of the trade-off, given the drastic increase in token count (e.g., from $(I/2)^2$ to $I^2$) and the associated computational cost surge. What are the authors' thoughts on this possibility?

**Question 5: Clarification on Patchify Implementation and Weight Initialization?**

Original DiT often implements Patchify using a Conv2D layer with kernel_size=stride=patch_size. However, Fig 3 and Sec 3.1 describe it as "extract patches + linear projection ($W_s$)". Are these formulations equivalent? If not, why choose the latter? Furthermore, could the authors clarify precisely how the weight initialization described in Sec 3.3 (averaging/duplicating) is applied to these independent linear layers $W_s$ and $W_s^u$ to avoid confusion?

---

> ### Author Response · Authors · 2025-11-24
> **Response to Reviewer R4cM (part1)**
>
> > **Response to Weakness 1**: Limited Novelty: While effective, the core idea (adjusting computation/token count based on noise level) is not entirely new. Related works like multi-scale/cascaded Diffusion, Pyramidal Flow (Jin et al., 2024), and especially concurrent works FlexiDiT (Anagnostidis et al., 2025) and Lumina-Video (Liu et al., 2025) explore the exact same core mechanism (time-varying patch sizes). PPFlow's primary novelty lies in its specific implementation (avoiding jump points via full-resolution operation) and its staged training strategy.
>
> We acknowledge that the high-level concept of adapting computation based on noise levels has been explored in prior works. However, PPFlow makes several distinct and novel contributions:
>
> - Full-resolution operation: Unlike Pyramidal Flow's multi-resolution approach, PPFlow operates consistently on full-resolution latent representations, eliminating discontinuities at resolution transitions.
> - Staged training: Our approach optimizes models specifically for different noise-level intervals with corresponding patch sizes, fundamentally different from full-timeline training in concurrent works (Tables 4-5 demonstrate superior performance).
> - Efficient initialization: PPFlow can be fine-tuned from pretrained models with minimal extra training cost (+7.1% FLOPs).
> - Comprehensive validation: We validate across multiple architectures (DiT, SiT, FLUX.1) and resolutions (256×256 to 2048×2048).
>
>
> > **Response to Weakness 2**: Insufficient Comparison with Concurrent Work: The paper mentions concurrent works FlexiDiT and Lumina-Video in Sec 2 and 4.4. Although it claims PPFlow's training strategy (staged training) is superior to their approach (full-timeline training) and experiments (Tables 4, 5) show better results for PPFlow, the argumentation for why PPFlow's strategy is better, and the trade-offs against the claimed "flexibility" of FlexiDiT, could be discussed more deeply.
>
> 1. The reason for why PPFlow's strategy is better.
>
> Forcing a model to handle both large-size and small-size patches at the same timestep potentially introduces conflicting optimization objectives. As discussed in (Hang et al., 2023), diffusion training is a multi-task learning problem; strategies like FlexiDiT increase the learning complexity.
>
> PPFlow avoids this issue. By design, **each stage focuses on a specific noise range, making the optimization task much simpler and allowing the model to learn better**. To validate this, we compare the training loss curves of PPFlow against the FlexiDiT strategy. As shown in Figure 9 in  Appendix of revised version, PPFlow achieves a consistently lower loss, indicating superior convergence.
>
> 2. Trade-off discussion.
>
> We acknowledge that FlexiDiT's full-timeline training offers the flexibility to adopt arbitrary inference schedules after training, which reduce training costs. While, identifying the optimal inference schedule among infinite possibilities remains non-trivial. In contrast, PPFlow ensures train-test consistency, reducing the cost of searching for the optimal inference schedule, and achieves better performance.
>
> Reference:
> [1] Hang, Tiankai, et al. "Efficient diffusion training via min-snr weighting strategy." Proceedings of the IEEE/CVF international conference on computer vision. 2023.
>
> > **Response to Weakness 3**: Arbitrary Stage Divisions and Patch Sizes: The timestep points for stage transitions (e.g., 0.5, 0.75) and the patch sizes used (e.g., 4x4, 4x2, 2x2) are manually chosen. The paper lacks sensitivity analysis on these hyperparameters or exploration of more automated selection methods. Increasing the number of stages (Table 7) seems to have diminishing returns and requires more training.
>
> 1. Sensitivity analysis.
>
> To address this, we conduct additional experiments on PPF-B-2 by training models with varying stage transitions. As shown in the table, all PPF-B-2 variants consistently outperform the baseline SiT-B/2 in both FID and IS metrics, regardless of the specific stage division used. This demonstrates that our method is **robust** to the choice of stage transitions.
>
>
> | Method   | Stage time points | FID-50K ↓ | IS ↑   |
> |----------|-------------------|-----------|--------|
> | SіT-B/2  | --                | 4.46      | 180.95 |
> | PPF-B-2  | [0.25, 1.0]       | 4.25      | 249.12 |
> |          | [0.4, 1.0]        | 4.27      | 250.30 |
> |          | [0.5, 1.0]        | 4.22      | 252.10 |
> |          | [0.6, 1.0]        | 4.31      | 251.28 |
> |          | [0.75, 1.0]       | 4.40      | 242.10 |
>
> 2. More automated selection methods.
>
> Following your suggestion, we will consider exploring automated interval selection based on metrics such as signal-to-noise ratio (SNR).
>
> 3. Question about diminishing returns and requires more training.
>
> We acknowledge that aggressive token reduction (adding more pyramid stages) does slightly affect generation quality, but it remains comparable with the baseline with substantial inference speedup.

---

> ### Author Response · Authors · 2025-11-24
> **Response to Reviewer R4cM (part2)**
>
> > **Response to Weakness 4**: Missing Key Visualizations: While Table 3 presents metric results for applying PPFlow to FLUX.1 (text-to-image model), the paper lacks corresponding visual samples. Given the subjective nature of text-to-image tasks, adding visual comparisons between PPF-FLUX.1 and the baseline FLUX.1-ft would significantly strengthen the conclusions.
>
> We thank the reviewer for the  suggestion. We demonstrate the visualization results for the origin FLUX.1, FLUX.1-ft and PPF-FLUX.1 using  prompts randomly from DPG-bench in Figure 7 and Figure 8 in the Appendix. The side-by-side comparisons across various text prompts shows that PPF-FLUX.1 maintains comparable visual quality to the baseline.
>
> > **Response to Question 1**: What are the fundamental trade-offs between PPFlow's staged training and FlexiDiT/Lumina-Video's full-timeline training?
> The paper suggests PPFlow's staged training allows models to better specialize for specific noise levels, and experiments show better performance than a re-implementation of the Lumina-Video approach. However, FlexiDiT claims their full-timeline training offers more "flexibility," potentially allowing the use of any patch size at any timestep during inference. Could the authors elaborate on the trade-offs? Is PPFlow's specialized approach inherently better under fixed schedules, while FlexiDiT's generalist approach offers more adaptability if the inference schedule (patch sizes vs. timesteps) needs modification? A direct experimental comparison of these training strategies (e.g., using PPFlow's architecture but FlexiDiT's training method) would be valuable.
>
> Please see our response to Weakness 2.
>
> > **Response to Question 2**:  How sensitive are the results to the choice of stage divisions (timesteps) and patch size combinations? The current stage divisions and patch sizes are set manually. How much do the final speedup and image quality depend on these specific choices? Are there better combinations? Could these parameters potentially be learned?
>
> Please see our response to Weakness 3.
>
> > **Response to Question 3**: How does PPFlow perform on tasks requiring fine spatial structure or long-range dependencies?
> Does using larger patches (which discard early high-frequency information) in high-noise stages negatively impact tasks that require precise spatial layout or fine-grained textures (e.g., complex scenes, generating images with text)? While ImageNet and T2I benchmark results are strong, how does performance hold up in more challenging scenarios? Suggest adding relevant experiments or discussion.
>
> This is an insightful question. To investigate the impact on fine spatial structures, we conduct additional experiments on LongText Bench and Spatial (T2I-CompBench sub-metric). The results indicate that PPF-FLUX.1 incurs **no performance degradation** compared to baselines.
>
> This is attributed to our architectural design: (1) The number of textual tokens is kept fixed throughout the process, ensuring no loss of long-range semantic information; (2) The visual tokens are pyramidally increasing, meaning the model operates on the full visual tokens during the final stage. This guarantees that high-frequency details and precise spatial layouts are fully recovered.
>
> | Method       | LongText Bench ↑ | Spatial (T2l-compbench) ↑ |
> |--------------|------------------|---------------------------|
> | FLUX.1       | 0.306            | 0.2870                    |
> | FLUX.1-ft    | 0.308            | 0.2875                    |
> | PPF-FLUX.1   | 0.306            | 0.2878                    |
>
> > **Response to Question 3**: Question 4: Could 1x1 patch sizes be considered for the final, very low-noise stages?
> To potentially capture the finest details in the last stages (close to t=1), would incorporating a 1x1 patch size (equivalent to operating directly on pixels/tokens) be beneficial? This might further improve image quality but would require careful consideration of the trade-off, given the drastic increase in token count  and the associated computational cost surge. What are the authors' thoughts on this possibility?
>
> We thank the reviewer for the insightful suggestion. Specifically, we modify the time segmentation from the original  $[0.5, 1.0]$ to $[0.5, 0.8, 1.0]$. In this new last stage $t \in [0.8, 1.0]$, we utilize a $1 \times 1$ patch size. The results are summarized below. As shown, incorporating the $1 \times 1$ patch size yields a performance boost.
>
> This validates the reviewer's hypothesis that capturing fine details  during the final generation stage is beneficial. This potentially helps generate high-fidelity textures and will be included in future work.
>
> | Method          | Stage time points   | FID ↓ | IS ↑   |
> |-----------------|---------------------|-------|--------|
> | SіT-B/2         | --                  | 4.46  | 180.95 |
> | PPF-B-2         | [0.5, 1.0]          | 4.22  | 252.10 |
> | + 1x1 patch size | [0.5, 0.8, 1.0]     | 4.01  | 260.22 |

---

> ### Author Response · Authors · 2025-11-24
> **Response to Reviewer R4cM (part3)**
>
> > **Response to Question 5**: Clarification on Patchify Implementation and Weight Initialization?
> Original DiT often implements Patchify using a Conv2D layer with kernel_size=stride=patch_size. However, Fig 3 and Sec 3.1 describe it as "extract patches + linear projection ($W_s$)". Are these formulations equivalent? If not, why choose the latter? Furthermore, could the authors clarify precisely how the weight initialization described in Sec 3.3 (averaging/duplicating) is applied to these independent linear layers $W_s$ and $W_s^u$ to avoid confusion?
>
> Yes, Conv2D with `kernel_size=stride=patch_size` is mathematically equivalent to "extract patches + linear projection". Both operations perform the same computation: reshaping spatial dimensions into channels and applying linear transformation. We describe it as "extract + project" for conceptual clarity.
>
> Regarding weight initialization, we adapt pre-trained weights to larger patch sizes to preserve the initial feature distribution. Specifically, we initialize the Patchify weights by concatenating and scaling the original weights ( $\mathbf{W}_2 = \frac{1}{4}[\mathbf{W}, \mathbf{W}, \mathbf{W}, \mathbf{W}]$), while the Unpatchify weights are initialized by duplicating the original weights to distribute the output signal uniformly. We add the related pytorch code in the Appendix (Listing 1) of revised version to fully clarify this process.

---

> ### Author Response · Authors · 2025-11-27
>
> Dear Reviewer R4cM,
>
> We sincerely appreciate your constructive feedback. We hope that our revisions and responses have  addressed your concerns, particularly regarding the **comparison with concurrent work**, the ablation studies on **stage divisions and patch sizes**, as well as the **visualizations and additional metrics** for text-to-image PPF-FLUX.1.
>
> As the discussion phase is drawing to a close, we would greatly value any further feedback. We remain available to clarify any pending issues and respectfully hope that you might reconsider your rating based on these updates. Many thanks!

---

### Official Review · Reviewer_SsjJ · 2025-10-28

**Soundness:** 3
**Presentation:** 3
**Contribution:** 2
**Rating:** 4
**Confidence:** 3

**Summary:**

The authors propose Pyramidal Patchification Flow (PPFlow), an efficient diffusion model that reduces the number of tokens at high-noise timesteps to accelerate generation and potentially also training. Instead of using a fixed patch size, PPFlow uses a multi-stage patch size schedule, which uses larger patches for earlier timesteps and smaller patches for later timesteps. All denoising transformer blocks are shared across stages. Thanks to this design, PPFlow speeds up inference by at least $1.6\times$ without compromising image generation quality.

**Strengths:**

- Speedup without compromising quality: PPFlow achieves $1.6\times - 2.0\times$ faster inference on image generation tasks while maintaining comparable image generation quality.
- Thanks to its design, all denoising steps happen in the same resolution, and it naturally resolves the "jump points" issues introduced in prior works. This leads to a simpler sampling scheme, which does not require tricks like renoising but also leads to a better image generation quality.
- Extensive experiments demonstrate the effectiveness of the method.

**Weaknesses:**

- Varying CFG scales: My primary concern lies in attributing the reported performance improvements to architectural innovations rather than to the varying classifier-free guidance (CFG) scales applied across different intervals. As stated in lines 357–359, the model adopts distinct CFG levels at different denoising stages. While this is a reasonable design choice for PPFlow, it closely parallels the Guidance Interval technique [1], which argues that guidance should not be applied uniformly throughout the denoising process since guidance at earlier timesteps can negatively impact sample diversity and quality. It would strengthen the paper if the authors could provide a direct comparison with Guidance Interval applied to SiTs to confirm that the observed quality gains are not merely due to this scheduling strategy.

- Diminishing returns with more stages: While adding more pyramid stages can further cut computation, it also compromises image generation quality. This seems like a trade-off to me, where aggressive token reduction demands more training but also negatively impacts the generation quality.

---
[1] Kynkäänniemi, T., Aittala, M., Karras, T., Laine, S., Aila, T., & Lehtinen, J. (2024). Applying guidance in a limited interval improves sample and distribution quality in diffusion models. Advances in Neural Information Processing Systems, 37, 122458-122483.

**Questions:**

- Regarding lines 280–283, are the time intervals empirically determined? It would be interesting to explore whether adaptive interval selection, based on certain metrics such as the signal-to-noise ratio (SNR), could further improve performance or stability.
- The choice of a $4\times 2$ patch configuration seems interesting, as most patch-based designs adopt square-shaped patches. Have the authors experimented with larger square configurations, such as $8\times8$ patches, especially for higher-resolution synthesis? Such an ablation could clarify whether the asymmetric patching contributes meaningfully to the model’s efficiency or generative quality.

---

> ### Author Response · Authors · 2025-11-24
> **Response to Reviewer SsjJ**
>
> > **Response to Weakness 1**: Varying CFG scales: My primary concern lies in attributing the reported performance improvements to architectural innovations rather than to the varying classifier-free guidance (CFG) scales applied across different intervals. As stated in lines 357–359, the model adopts distinct CFG levels at different denoising stages. While this is a reasonable design choice for PPFlow, it closely parallels the Guidance Interval technique [1], which argues that guidance should not be applied uniformly throughout the denoising process since guidance at earlier timesteps can negatively impact sample diversity and quality. It would strengthen the paper if the authors could provide a direct comparison with Guidance Interval applied to SiTs to confirm that the observed quality gains are not merely due to this scheduling strategy.
>
> This is an important point that deserves clarification. To address this, we conducted a direct comparison between our PPF-B-2 and the SiT-B/2 baseline under same guidance settings. We evaluate two specific configurations: 1) Fixed CFG: employing a constant scale of 1.5 uniformly throughout the denoising process, consistent with the original SiT paper; and 2) Guidance Interval: applying a guidance scale of 2.0 exclusively within the timestep interval $t \in [0.1, 0.9]$, guidance scale follows the parameters suggested in Guidance Interval paper.
>
> **Our PPF-B-2 consistently outperforms the SiT-B/2 baseline under both the Fixed CFG and Guidance Interval configurations**. The results confirm that performance gains are primarily driven by our architectural innovations rather than the varying CFG.
>
> | Method       | Guidance        | FID ↓ | IS ↑   | Testing FLOPs |
> |--------------|-----------------|-------|--------|---------------|
> | SiT-B/2      | Fixed CFG       | 4.46  | 180.95 | 100%          |
> |              | Guidance Interval | 4.37  | 169.56 | 90.0%         |
> | PPF-B-2      | Fixed CFG       | 4.44  | 201.12 | 62.0%         |
> |              | Guidance Interval | 4.25  | 194.70 | 59.6%         |
>
> > **Response to Weakness 2**: Diminishing returns with more stages: While adding more pyramid stages can further cut computation, it also compromises image generation quality. This seems like a trade-off to me, where aggressive token reduction demands more training but also negatively impacts the generation quality.
>
> We acknowledge that aggressive token reduction (adding more pyramid stages) does slightly affect generation quality, but it remains comparable with the baseline with substantial inference speedup.
>
> > **Response to Question 1**: Regarding lines 280–283, are the time intervals empirically determined? It would be interesting to explore whether adaptive interval selection, based on certain metrics such as the signal-to-noise ratio (SNR), could further improve performance or stability.
>
> Yes, the time intervals in our experiments are empirically determined. Following your suggestion, we will consider exploring adaptive interval selection based on signal-to-noise ratio (SNR) in the future work.
>
> > **Response to Question 2**: The choice of a $4 \times 2$ patch configuration seems interesting, as most patch-based designs adopt square-shaped patches. Have the authors experimented with larger square configurations, such as $8 \times 8$patches, especially for higher-resolution synthesis? Such an ablation could clarify whether the asymmetric patching contributes meaningfully to the model’s efficiency or generative quality.
>
>
> We thank the reviewer for the insightful suggestion. To verify the effectiveness of our design, we conduct an  ablation study under the same testing FLOPs. We compare two configurations of the PPF-B-3 model: a symmetric configuration utilizing $8\times8 \to 4\times4 \to 2\times2$ patch size, and our proposed asymmetric configuration which employs a schedule of $4\times4 \to 4\times2 \to 2\times2$.
>
>
> As shown in the table, the symmetric scheme requires more training  to match the performance of our asymmetric design at 1M steps. This suggests that smoother asymmetric transition is necessary for training efficiency.
>
>
> | Method   | Pyramid patchsize                                   | Testing FLOPs | Training steps | FID-50k↓ | IS ↑   |
> |----------|-----------------------------------------------------|---------------|----------------|-------|--------|
> | SiT-B/2  | --                                                  | 100%          | --             | 4.46  | 180.95 |
> | PPF-B-3  | $ 4 \times 4 \rightarrow 4 \times 2 \rightarrow 2 \times 2 $ (asymmetric) | 49%           | 1M             | 4.57  | 236.53 |
> |          | $ 8 \times 8 \rightarrow 4 \times 4 \rightarrow 2 \times 2 $ (symmetric)  | 49%           | 1M             | 4.65  | 232.78 |
> |          | $ 8 \times 8 \rightarrow 4 \times 4 \rightarrow 2 \times 2 $ (symmetric)  | 49%           | 1.2M           | 4.55  | 236.14 |

---

> ### Author Response · Authors · 2025-11-27
>
> Dear Reviewer SsjJ,
>
> Thank you for your helpful feedback. We hope that our latest revisions and responses effectively resolve your concerns, specifically by providing comparisons with **same CFG scales and guidance intervals**, and by including the requested ablation study on **asymmetric patch sizes**.
>
> We are happy to answer any final questions before the discussion window closes. If you find our response satisfactory, we would be grateful if you could consider raising your score. Thanks for your time!

---

### Official Review · Reviewer_wyzG · 2025-10-28

**Soundness:** 2
**Presentation:** 3
**Contribution:** 2
**Rating:** 4
**Confidence:** 4

**Summary:**

The paper explores adaptive patch sizes for DiT models trained for image generation. In contrast to prior works, it argues for separating the patch sizes for different time segments to better align with the optimal test-time inference strategy. It tests on top of SiT for class-conditional image generation on ImageNet and Flux-dev for text-to-image generation. It reports better qualitative results for both setups.

**Strengths:**

- The paper presents several useful ablations for adaptive patch size strategies: patch-level embedding, stage-wise CFG,
- The paper is written well
- Overall, this paper looks like a useful reference for adaptive patch size exploration

**Weaknesses:**

- FLOPs are not a good metric to measure training speed. Clock wall time should be reported as well.
- Figure 3 is not too informative of the patchification process since it presents the input and output the same. Also, the input to patchification is a 2D image patch, not flattenning is a part of patchification.
- Typo on L248: [WWWW] reads like a matrix multiplication of 4 matrices W. Same for L252
- The paper claims to train Flux-dev, but does not show a single generated image from it. GenEval is not a reliable benchmark and cannot be the only source of assessment.
- The authors claim that Lumina-Video's strategy to use all the patch sizes for all the timesteps is inferior. But it's unclear if its cfg is just as tuned and if the proposed strategy supports autoguidance like Lumina-Video does.

**Questions:**

- Is it possible to support autoguidance (using a large-patch model as the weak one)?

---

> ### Author Response · Authors · 2025-11-24
> **Response to Review wyzG**
>
> > **Response to Weakness 1**: FLOPs are not a good metric to measure training speed. Clock wall time should be reported as well.
>
> We agree that clock wall time provides a practical measure of training efficiency. We measure the actual training time per iteration on NVIDIA H200 GPUs. The results are presented in the table below. Although the wall-clock time reduction is slightly smaller than the FLOPs reduction, it still yields a substantial speed advantage, **achieving a 32%–45% speedup**.
>
> | Method             | Clock wall Time ( s / iteration ) | Time Reduction |
> |--------------------|-----------------------------------|----------------|
> | SiT-B/2 (256)      | 0.062                             | -              |
> | PPF-B-2 (256)      | 0.042                             | 32.3%          |
> | PPF-B-3 (256)      | 0.034                             | 45.2%          |
> | SiT-XL/2 (256)     | 0.237                             | -              |
> | PPF-XL-2 (256)     | 0.161                             | 32.1%          |
> | PPF-XL-3 (256)     | 0.129                             | 45.6%          |
> | DiT-XL/2 (512)     | 1.538                             | -              |
> | PPF-XL-2 (512)     | 1.044                             | 32.1%          |
> | PPF-XL-3 (512)     | 0.837                             | 45.5%          |
>
> > **Response to Weakness 2**: Figure 3 is not too informative of the patchification process since it presents the input and output the same. Also, the input to patchification is a 2D image patch, not flattenning is a part of patchification.
>
> We thank the reviewer for the insightful suggestion. We update Figure 3 to better explain the linear projection within the patchification process.
>
> > **Response to Weakness 3**: Typo on L248: [WWWW] reads like a matrix multiplication of 4 matrices W. Same for L252
>
> We thank the reviewer for raising these typos. We update the notation throughout the paper. Specifically, we have added commas to explicitly denote vector concatenation.
>
> > **Response to Weakness 4**: The paper claims to train Flux-dev, but does not show a single generated image from it. GenEval is not a reliable benchmark and cannot be the only source of assessment.
>
> We thank the reviewer for the insightful suggestion. We add the following to address it:
> - **Visual results**: We now include Figure 7 and Figure 8 in the revised version showing qualitative comparisons of images generated by the  origin Flux.1 model,  the Flux.1-ft  model and our PPF-FLUX.1 model. The examples cover diverse prompts (from DPG-bench) and demonstrate that PPFlow maintains visual quality.
> - **Additional evaluation metrics**: Beyond GenEval, DPGbench and T2ICompbench, we  add the human preference score (HPS v2.1) for PPF-FLUX.1 model. These additional evaluations provide more comprehensive and reliable assessment of PPF-FLUX.1 model.
>
> | Method       | HPS v2.1 ↑ |
> |--------------|------------|
> | FLUX.1       | 30.52      |
> | FLUX.1-ft    | 30.60      |
> | PPF-FLUX.1   | 30.62      |
>
> > **Response to Weakness 5**: The authors claim that Lumina-Video's strategy to use all the patch sizes for all the timesteps is inferior. But it's unclear if its cfg is just as tuned and if the proposed strategy supports autoguidance like Lumina-Video does.
>
> We appreciate this important clarification request. To ensure a fair and rigorous comparison, we align the evaluation settings for both the Lumina-Video baseline and our PPF-B-2. Specifically, we evaluate two guidance strategies: 1) Fixed CFG: we use a scale of 1.5 uniformly throughout the denoising process, consistent with the original SiT paper setting. 2) Autoguidance: we adopt a scale of 2.05, following the protocol in Autoguidance paper. For the implementation of Autoguidance, we employ a SiT-B model with $8 \times 8$ patches to serve as the guiding model.
>
> The results in the table demonstrate that our **PPF is fully compatible with the Autoguidance**. Furthermore, **under these same settings, PPF-B-2 consistently outperforms the Lumina-Video method**.
>
> | Method       | Guidance     | FID-50k ↓ | IS ↑  |
> |--------------|--------------|-------|-------|
> | Lumina-Video | Fixed CFG    | 18.77 | 79.12 |
> |              | Autoguidance | 17.98 | 71.54 |
> | PPF-B-2      | Fixed CFG    | 16.88 | 82.09 |
> |              | Autoguidance | 16.13 | 75.18 |
>
> > **Response to Question 1**: Is it possible to support autoguidance (using a large-patch model as the weak one)?
>
> Please see our response to Weakness 5.

---

> ### Author Response · Authors · 2025-11-27
>
> Dear Reviewer wyzG,
>
> We sincerely appreciate your constructive feedback. We have carefully revised the manuscript to address your concerns, particularly regarding the **wall-clock time comparison, figure updates, and typo corrections**. Furthermore, we have added the **visualization results and human preference scores** for the text-to-image model PPF-FLUX.1, along with the **autoguidance comparison** against Lumina-Video.
>
> We would value any further feedback before the discussion phase concludes and remain available to answer any final questions. We respectfully hope that you might consider raising your rating based on these improvements. Thanks!

---

### Official Review · Reviewer_uvnV · 2025-10-30

**Soundness:** 3
**Presentation:** 3
**Contribution:** 3
**Rating:** 6
**Confidence:** 4

**Summary:**

This paper proposes Pyramidal Patchification Flow (PPFlow), a method that dynamically varies the patch size of DiT across timesteps—using larger patches in higher-noise timesteps to reduce token count and smaller patches in lower-noise timesteps to preserve visual fidelity. PPFlow operates on the full-resolution representation throughout the generation process and trains the model for each patch size only over the corresponding timestep range. As a result, PPFlow achieves faster inference without generation quality degradation.

**Strengths:**

- The paper is well-structured, providing clear explanations of design choices and experimental results. It includes comprehensive evaluations and ablation studies that effectively validate the proposed approach.
- The proposed PPFlow patchification strategy can be applied to various transformer-based diffusion models due to its simplicity and comparable generation quality to conventional fixed patch-size models.

**Weaknesses:**

- There are several typographical errors in the paper. For example, there are double commas in the abstract (line 36, p1); in line 222 (p.5), the word “depent” should be “dependent”; and in Table 5 (p.8), “pyrimid rep” appears to be a typo.

**Questions:**

- In Table 7, the results show that increasing the number of training steps helps recover generation quality (line 445 vs 446 & line 447 vs 448). In Table 2, instead of matching the number of training steps, what happens if we match the training FLOPs between the two-level and three-level methods? How does the generation quality compare under this training-cost-matched condition?

---

> ### Author Response · Authors · 2025-11-24
> **Response to Review uvnV**
>
> > **Response to Weakness 1**: There are several typographical errors in the paper. For example, there are double commas in the abstract (line 36, p1); in line 222 (p.5), the word “depent” should be “dependent”; and in Table 5 (p.8), “pyrimid rep” appears to be a typo.
>
> We thank the reviewer for raising these typographical errors. We carefully proofread the entire manuscript and correct all identified typos throughout the paper.
>
> > **Response to Question 1**: In Table 7, the results show that increasing the number of training steps helps recover generation quality (line 445 vs 446 & line 447 vs 448). In Table 2, instead of matching the number of training steps, what happens if we match the training FLOPs between the two-level and three-level methods? How does the generation quality compare under this training-cost-matched condition?
>
> We thank the reviewer for the insightful suggestion. We conduct additional experiments to compare two-level and three-level PPFlow in Table 2 under same training FLOPs  cost. Specifically, we adjusted the training steps for the three-level method (PPF-3) to match the FLOPs of the two-level method (PPF-2). The results are presented in the table below. With matched training FLOPs, the **three-level method surpasses the baseline and reduces the gap with the two-level method**. We will add these results to the revised version.
>
> | Method     | Size | Training Steps | Training FLOPs (%) | FID-50k ↓ | IS ↑   |
> |------------|------|----------------|---------------------|-----------|--------|
> | SiT-B/2    | 256  | --             | 100                 | 4.46      | 180.95 |
> | PPF-B-2    | 256  | 1M             | 8.9                 | 4.22      | 252.10 |
> | PPF-B-3    | 256  | 1M             | 7.1                 | 4.57      | 236.53 |
> |            |      | 1.25M          | 8.9                 | 4.28      | 248.13 |
> | SiT-XL/2   | 256  | --             | 100                 | 2.15      | 258.09 |
> | PPF-XL-2   | 256  | 1M             | 8.9                 | 1.99      | 271.62 |
> | PPF-XL-3   | 256  | 1M             | 7.1                 | 2.23      | 286.67 |
> |            |      | 1.25M          | 8.9                 | 2.09      | 289.17 |
> | DiT-XL/2   | 512  | --             | 100                 | 3.04      | 240.82 |
> | PPF-XL-2   | 512  | 400k           | 7.6                 | 3.01      | 249.98 |
> | PPF-XL-3   | 512  | 400k           | 5.8                 | 3.06      | 249.91 |
> |            |      | 525k           | 7.6                 | 3.03      | 250.64 |

---

> ### Author Response · Authors · 2025-11-27
>
> Dear Reviewer uvnV,
>
> Thank you for your helpful comments. We trust that our  response effectively addresses your points, specifically regarding the performance analysis of the three-level further training.
>
> We remain available for any final clarifications before the discussion period ends. Thank you for your time and guidance!

---

### Author Response · Authors · 2025-12-03
**Summary of Rebuttal and Discussion Phase (Part1)**

Dear Area Chair,

We sincerely thank the reviewers for their time, thoughtful feedback, and voluntary service, and we greatly appreciate your coordination in overseeing our submission.

We are deeply encouraged by the reviewers’ praise and commendations. Below we summarize their **positive** feedback:

1. **Simple and Intuitive**. (Reviewers R4cM and uvnV)

   Reviewers commend the simplicity and intuitiveness of our approach. Reviewer R4cM acknowledges that "The core idea is intuitive, simple, and easy to implement." Similarly, Reviewer uvnV notes that the strategy "can be applied to various transformer-based diffusion models due to its simplicity."

2. **Substantial Inference Speedups and Comprehensive Experiments.** (Reviewers SsjJ, R4cM and uvnV)

   The reviewers acknowledge the method's efficiency and the thoroughness of our evaluation. Reviewer SsjJ emphasizes that "PPFlow achieves $1.6 - 2.0 \times $ faster inference on image generation tasks while maintaining comparable image generation quality," a result Reviewer R4cM points out "Experimental results (Tables 1, 2, 3) clearly demonstrate that PPFlow achieves substantial inference speedups  across multiple models (SiT-B, SiT-XL, DiT-XL, FLUX.1) and resolutions (256 to 2048)." Reviewer uvnV further praise the "comprehensive evaluations and ablation studies that effectively validate the proposed approach."

3. **Avoiding the "Jump Point" Issue**. (Reviewers SsjJ, R4cM and wyzG)

   Reviewers positively note that our design effectively solves resolution inconsistencies. Reviewer SsjJ observe that "all denoising steps happen in the same resolution," which "naturally resolves the 'jump points' issues introduced in prior works." This is echoed by Reviewer R4cM, who states that operating on full-resolution latents "avoids the complexities associated with 'jump points' and re-noising sampling strategies, keeping the inference process simple," while Reviewer wyzG regard the work as "a useful reference for adaptive patch size exploration."



We acknowledge the reviewers' concerns regarding guidance technique impacts (wyzG, SsjJ) and novelty in the high-level idea--adjusting token count based on noise level--is not entirely new (R4cM). We address the former with a comprehensive comparison of guidance strategies (Points 1-2), demonstrating superior performance. For the latter, we clarity our technical contributions and demonstrate superior performance in training loss and generation quality versus prior works (Points 3-4). We sincerely hope the Area Chair will take these revisions and clarifications into consideration for a positive assessment.

During the response phase, we directly addressed the concerns and clarified misunderstandings raised by the reviewers:

1. **Autoguidance support and comparison with Lumina-Video**. (Reviewer wyzG)

   Our method can support autoguidance. Compared to Lumina-Video, our method yields superior generation quality under fixed-scale CFG and autoguidance. Specifically, we achieve an FID of 16.88 (vs. 18.77) and IS of 82.09 (vs. 79.12) under fixed CFG, and an FID of 16.13 (vs. 17.98) and IS of 75.18 (vs. 71.54) with autoguidance.

2. **A direct comparison with Guidance Interval applied to baseline**. (Reviewer SsjJ)

   Compared to the baseline model, our method achieves better quality under fixed-scale CFG (FID: 4.46 $\rightarrow$ 4.44, IS: 180.95 $\rightarrow$ 201.12)  and Guidance Intervals (FID: 4.37 $\rightarrow$ 4.25, IS: 169.56 $\rightarrow$ 194.70).

3. **Limited Novelty**.  (Reviewer R4cM)

   We acknowledge that the high-level idea--adjusting token count based on noise level--is not entirely new. While we adopt a different implementation toward this goal. Specifically, PPFlow makes several distinct and novel contributions:

- Full-resolution operation: Unlike Pyramidal Flow's multi-resolution approach, PPFlow operates consistently on full-resolution latent representations, eliminating discontinuities at resolution transitions.
- Staged training: Our approach optimizes models specifically for different noise-level intervals with corresponding patch sizes, fundamentally different from full-timeline training in concurrent works (Tables 4-5 demonstrate superior performance).
- Efficient initialization: PPFlow can be fine-tuned from pretrained models with minimal extra training cost (+7.1% FLOPs).

4. **More training strategies comparison with concurrent works FlexiDiT/Lumina-Video**. (Reviewer R4cM)

   We report the training loss curves for our method compared to FlexiDiT and Lumina-Video in Appendix Figure 9, demonstrating that PPFlow achieves a consistently lower and stable loss ( 0.763 vs. 0.774), indicating superior convergence. By focusing each stage on a specific noise range, the optimization task of PPFlow becomes much simpler, allowing the model to learn more effectively.

---

> ### Author Response · Authors · 2025-12-03
> **Summary of Rebuttal and Discussion Phase (Part2)**
>
> 5. **Performance comparison with matched training FLOPs between the two-level and three-level methods**. (Reviewer uvnV)
>
>    To address it, we adjusted the training steps for the three-level method (PPF-3) to match the FLOPs of the two-level method (PPF-2). PPF-3 surpasses the baselines (SiT-B, SiT-XL, DiT-XL) with FIDs of 4.28, 2.09, and 3.03, while narrowing the FID performance gaps with PPF-2 to only 0.06, 0.10, and 0.02, respectively.
>
> 6. **Clock wall time for training**. (Reviewer wyzG)
>
>    We also report the wall-clock training time; our method achieves a 32%-45% reduction in wall-clock time per training iteration.
>
> 7. **Visualization results and extra metrics for PPF-FLUX.1**. (Reviewer wyzG, R4cM)
>
>    In the revised version, we add visualization results of a variant of our method based on FLUX.1-dev, named PPF-FLUX.1 (see Appendix Figures 7 and 8). It demonstrates that high generation quality and details are maintained. Quantitative evaluations also validate our approach: HPS v2.1 (30.62 vs. 30.60), Longtext Bench (0.306 vs. 0.308), and spatial metrics (0.2878 vs. 0.2875). Results (ours vs. baseline) indicate performance that is consistently comparable to or better than the baseline.
>
> 8. **The effect of asymmetric patches**. (Reviewer SsjJ)
>
>    Regarding the effect of asymmetric patches, we find that under the same testing FLOPs, using an asymmetric patch size between two symmetric patchification sizes in the timestep improves the model's training convergence efficiency by 20%.
>
> 9. **Sensitive analysis of stage divisions.** (Reviewer R4cM)
>
>    We conduct additional experiments on PPF-B-2 by training models with varying stage transitions. Specifically, the models trained with varying stage points yield FIDs of 4.25, 4.27, 4.22, 4.31, and 4.40. The corresponding IS scores of 249.12, 250.30, 252.10, 251.28, and 242.10. All PPF-B-2 variants consistently outperform the baseline SiT-B/2  in both FID (4.46) and IS (180.95) metrics, demonstrating that the final performance is insensitive to stage divisions.
>
> 10. **Incorporating $1 \times 1$ patchification**. (Reviewer R4cM)
>
>     In very low-noise stages, adopting $1 \times 1$ patches in low-noise stages enhances fine detail preservation, boosting performance from 4.22 to 4.01 FID.
>
> 11. **Clarification on patchify implementation and weight initialization**.  (Reviewer R4cM)
>
>     Regarding the clarification on patchify implementation and weight initialization, we add explanations and include relevant code in Listing 1 of the Appendix in the revised version.
>
>
>
> Thank you very much for your time and consideration. We would be grateful if you could take into account the progress made during the rebuttal.
>
> Sincerely,
>
> The authors

---

### Meta-Review · Area_Chair_dyqg · 2026-01-07

**Summary:**

This paper proposes Pyramidal Patchification Flow (PPFlow), which improves diffusion transformer efficiency by using larger patches at high-noise timesteps and smaller patches at low-noise timesteps. Unlike Pyramidal Flow, PPFlow operates on full-resolution latent representations, avoiding "jump points" and re-noising tricks. The method achieves ~2× inference speedup while maintaining generation quality across multiple architectures (SiT, DiT, FLUX.1).

Initial scores were 6/4/4/4. Authors provided comprehensive responses with new experiments addressing all concerns. No reviewers engaged during discussion.

**Reviewer Concerns:**

Concerns successfully addressed:
- Guidance technique attribution (SsjJ, wyzG).
Reviewers questioned whether improvements came from architectural innovation or varying CFG scales, and also whether it supports autoguidance. Authors provided definitive comparison under identical guidance settings: fixed CFG, guidance interval and autoguidance, and demonstrated the gains stem from architecture, not guidance strategy.
- Limited novelty (R4cM).
Authors clarified distinct contributions: 1) Full-resolution operation (vs. Pyramidal Flow's multi-resolution). 2) Staged training (vs. full-timeline training in FlexiDiT/Lumina-Video). 3) Efficient fine-tuning from pretrained models (+7.1% FLOPs).
- Visualization results and extra metrics for PPF-FLUX.1 (wyzG, R4cM).
Authors added visualization results of a variant of our method based on FLUX.1-dev, named PPF-FLUX.1 and quantitative evaluations to validate the proposed approach: HPS v2.1 (30.62 vs. 30.60), Longtext Bench (0.306 vs. 0.308), and spatial metrics (0.2878 vs. 0.2875).
- Three-level vs two-level matched FLOPs (uvnV).
Authors adjusted training steps to match FLOPs, showing PPF-3 surpasses baselines (FID 4.28 vs 4.46 for SiT-B) while narrowing gap with PPF-2 (4.28 vs 4.22).
- Sensitivity to stage divisions (R4cM).
Authors provided ablation across different stage transition points, showing robust performance and demonstrating the proposed method is insensitive to specific stage choices.
- Asymmetric patch effectiveness (SsjJ).
Authors demonstrated asymmetric patches (4×2) improve convergence efficiency by 20% over symmetric alternatives under same FLOPs.

Outstanding Concerns:
I think the rebuttal well addressed the major concerns. One note: a few results in the rebuttal (e.g., quantitative evaluations of PPF-FLUX.1) are not incorporated into the revised paper. I encourage the authors to add them in the next version of the paper.

**Reviewer Scores:**

- Reviewer uvnV: Remains at 6.
- Reviewer wyzG: Likely rises to 6 (from 4).
- Reviewer SsjJ: Likely rises to 6 (from 4).
- Reviewer R4cM: Likely rises to 6 (from 4).

---

### Decision · Program_Chairs · 2026-01-26

Accept (Poster)